

# Fast responses on pre-industrial climate from present-day aerosols in a CMIP6 multi-model study

Prodromos Zanis[1*], Dimitris Akritidis[1], Aristeidis K. Georgoulias[1], Robert J. Allen[2], Susanne E. Bauer[3], Olivier Boucher[4], Jason Cole[5], Ben Johnson[6], Makoto Deushi[7], Martine Michou[8], Jane Mulcahy[6], Pierre Nabat[8], Dirk Olivie[9], Naga Oshima[7], Adriana Sima[4], Michael Schulz[9], Toshihiko Takemura[10]

[1]Department of Meteorology and Climatology, School of Geology, Aristotle University of Thessaloniki, Thessaloniki, Greece

[2]Department of Earth Sciences, University of California Riverside, Irvine, USA

[3]NASA Goddard Institute for Space Studies, New York, USA

[4]CNRS, LMD/IPSL, Sorbonne Université, Paris, France

[5]Environment and Climate Change Canada, Toronto, Canada

[6]Met Office, Exeter, UK

[7]Meteorological Research Institute, Japan Meteorological Agency, Tsukuba, Japan

[8]CNRM, Université de Toulouse, Météo-France, CNRS, Toulouse, France

[9]Norwegian Meteorological Institute, Oslo, Norway

[10]Research Institute for Applied Mechanics, Kyushu University, Fukuoka, Japan

*Correspondence to*: Prodromos Zanis (zanis@geo.auth.gr)

**Abstract.** In this work, we use Coupled Model Intercomparison Project Phase 6 (CMIP6) simulations from 10 Earth System Models (ESMs) and General Circulation Models (GCMs) to study the fast climate responses on pre-industrial climate, due to present-day aerosols. All models carried out two sets of simulations; a control experiment with all forcings set to the year 1850 and a perturbation experiment with all forcings identical to the control, except for aerosols with precursor emissions set to the year 2014. In response to the pattern of all aerosols effective radiative forcing (ERF), the fast temperature responses are characterised by cooling over the continental areas, especially in the Northern Hemisphere, with the largest cooling over East Asia and India, sulfate being the dominant aerosol surface temperature driver for present-day emissions. In the Arctic there is a warming signal for winter in the ensemble mean of fast temperature responses, but the model-to-model variability is large, and it is presumably linked to aerosol induced circulation changes. The largest fast precipitation responses are seen in the tropical belt regions, generally characterized by a reduction over continental regions and a southward shift of the tropical rain belt. This is a characteristic and robust feature among most models in this study, associated with a southward shift of the Intertropical convergence zone (ITCZ) and a weakening of the monsoon systems around the globe (Asia, Africa and America) in response to hemispherically asymmetric cooling from a Northern Hemisphere aerosol perturbation, leading the ITCZ and tropical precipitation to shift away from the cooled hemispheric pattern. An interesting feature in aerosol induced circulation changes is



a characteristic dipole pattern with intensification of the Icelandic Low and an anticyclonic anomaly over Southeastern Europe, inducing warm air advection towards the northern polar latitudes in winter.

# 1 Introduction

Aerosols interact directly with radiation through scattering and absorption (Haywood and Boucher, 2000) as well as with clouds by acting as cloud condensation nuclei (CCN) and ice nuclei (IN), affecting the Earth's radiation budget and climate (Lohmann and Feichter, 2005), while this impact can be much stronger on a regional scale (Ramanathan and Feng, 2009). On a global scale, aerosols have an inhomogeneous spatial distribution, due to their relatively short lifetime, closely following the patterns of regional emission sources. As a consequence, aerosols have a larger geographical variation in radiative forcing than $CO_2$, with the pattern and spatial gradients of their forcing affecting global and regional temperature responses as well as the hydrologic cycle and precipitation patterns (Myhre et al., 2013). In general, absorbing aerosols, like black carbon, tend to warm the climate and stabilize the atmosphere, while sulfate aerosols tend to cool the climate (Bond et al., 2013), but the aerosol induced circulation changes influence the spatial patterns of temperature and precipitation response to the regional aerosol forcing, while aerosol-cloud interactions complicate further these responses (Baker et al., 2015; Boucher et al., 2013; Rosenfeld et al., 2014b). While the local influence of aerosols close to their emmision sources has been clearly seen in a number of studies (Bartlett et al., 2018; Ramanathan and Feng, 2009; Sarangi et al., 2018; Thornhill et al., 2018; Zhang et al., 2018), their impact can extend beyond their emission regions via fast and slow climate responses (Andrews et al., 2010; Boucher et al., 2013; Kvalevåg et al., 2013). Reduction in sulphur emissions in China was found to lead to increases in temperature in much of the US, northern Eurasia, and the Arctic (Kasoar et al., 2016). Removal of U.S. anthropogenic $SO_2$ emissions showed robust patterns of temperature responses over land, with increases in temperature for most of the Northern Hemisphere land regions and the strongest response towards the Arctic (Conley et al., 2018; Shindell et al., 2015). Other recent model studies indicate an amplification of the temperature response towards the Arctic due to local and remote aerosol forcing (Stjern et al., 2017; Westervelt et al., 2018; Stjern et al., 2019). Furthermore, model perturbation simulations with increasing $SO_2$ in Europe, North America, East Asia and South Asia, showed a consistent cooling almost everywhere over the Northern Hemisphere with the Arctic revealing the largest temperature response in all experiments (Lewinschal et al., 2019). The investigation of temperature and precipitation responses to single-species forcings in different latitudinal bands showed that the influence of remote forcings on certain regions can often outweigh and even have an opposite sign to the influence of local forcings (Shindell et al., 2012).

The Intergovernmental Panel on Climate Change (IPCC) Fifth Assessment Report (AR5) has clarified the importance of distinguishing instantaneous radiative forcing and fast responses (through rapid atmospheric adjustments which modify the radiative budget indirectly) from slow responses through feedbacks (affecting climate variables that are mediated by a change in surface temperature) (Boucher et al., 2013). The dual fast response (or rapid adjustment) and slow response framework has been verified across a range of recent global model studies (Baker et al., 2015; Liu et al., 2018; Richardson.,



2015; Samset et al., 2016, 2018). Rapid adjustments affect cloud cover and other components of the climate system and thereby alter the global radiation budget indirectly within a few weeks, much faster than responses of the ocean to forcing (Myhre et al., 2013). Earlier studies have found that rapid adjustments have been important for global precipitation changes (Andrews et al. 2010; Kvalevåg et al. 2013) and regional temperature changes, but generally, the zonal means of slow
precipitation and temperature responses are stronger than the fast responses (Samset et al., 2016; Lewinschal et al., 2019; Baker et al., 2015; Stjern et al., 2017; Voigt et al., 2017). Under the framework of the Precipitation Driver Response Model Intercomparison Project (PDRMIP), multiple model results indicate that the global fast precipitation response to regional aerosol forcing scales with global atmospheric absorption, and the slow precipitation response scales with global surface temperature response (Liu et al., 2018). Another recent PDRMIP multi-model study showed that unlike other drivers of
climate change, the response of temperature and cloud profiles to the black carbon (BC) forcing is dominated by rapid adjustments causing weak surface temperature response to increased BC concentrations (Stjern et al., 2017). While some aspects of the regional variation in precipitation and temperature predicted by climate models appear robust, there is still a large degree of inter-model differences unaccounted for, because of uncertainties involved in the related modeling aspects, such as representation of aerosols, their vertical distribution and radiative properties, parameterizations of aerosol removal
processes including both wet and dry removal as well as aerosol-cloud interactions (Rosenfeld et al., 2014a; Shindell et al., 2015; Wilcox et al., 2015).

A forcing that accounts for rapid adjustments is termed as the effective radiative forcing (ERF) and conceptually represents the change in the net top of the atmosphere (TOA) radiative flux after allowing for atmospheric temperatures, water vapour and clouds to adjust, but with global mean surface temperature or a portion of surface conditions unchanged. A
standard method to investigate the fast responses in climate simulations to forcing from aerosols or other short lived climate forcers (SLCFs) is by fixing sea surface temperatures (SSTs) and sea ice cover (SIC) at climatological values, allowing all other parts of the system to respond until reaching steady state (Hansen et al., 2005). In this way, the climate response to a forcing agent in the fixed SST simulations is without any ocean response to climate change and therefore only weakly coupled to feedback processes through land surface responses (Myhre et al., 2013).

Here, we present a first analysis of the fast responses on pre-industrial climate due to present-day aerosols in a multi-model study based on simulations with 10 CMIP6 models. Section 2 presents the data used and the methodology applied in this study. In Section 3 the key results of this study are presented and discussed, while, finally, in Section 4 the main conclusions are summarized.

## 2 Data and methodology

In this work, we use CMIP6 simulations from 10 different models, namely CanESM5, CESM2, CNRM-CM6-1, CNRM-ESM2-1, GISS-E2-1-G, IPSL-CM6A-LR, MIROC6, MRI-ESM2-0, NorESM2-LM and UKESM1-0-LL. The aforementioned simulations were implemented within the framework of the Aerosol Chemistry Model Intercomparison Project (AerChemMIP),





which is endorsed by CMIP6 and aims at quantifying the impacts of aerosols and chemically reactive gases on climate and air quality (Collins et al., 2017). All models carried out two sets of simulations considering both aerosol-radiation and aerosol-cloud interactions: the piClim-control (with all forcings set to the year 1850 using aerosol precursor emissions of 1850) and the piClim-aer (again with all forcings set to 1850 but using aerosol precursor emissions of the year 2014). All simulations

covering at least a period of 30 years in total using fixed climatological average sea surface temperatures (SSTs) and sea ice distributions corresponding the year 1850. Furthermore, concentrations of well mixed greenhouse gases (WMGHGs), emissions of ozone precursors and ozone depleting halocarbons, solar irradiance forcing and land use are also set to the year 1850. The year 1850 is considered here as a pre-industrial period although it could be also assigned as an early industrial period. The perturbation experiments (e.g. piClim-aer) are run similarly for the 30 years period following the control

experiments (piClim-control), using the same control SST and sea ice, but with emissions for aerosol precursors set to present-day (2014) levels. It has to be noted that only 1 realization is analyzed for each model (see Table 1).

By subtracting the piClim-control simulations from the piClim-aer simulations the fast responses of pre-industrial climate to present-day aerosols are estimated since SST and sea ice distributions are fixed in the simulations. In this work, we examine the effect of aerosols on: 1) net radiative flux (shortwave and longwave) at the top of the atmosphere (TOA) which

manifests the ERF, 2) surface air temperature, 3) precipitation and 4) atmospheric circulation (wind and geopotential height at 850 hPa). As the horizontal resolution ranges between the different models (from 0.95º x 1.25º to 2.8º x 2.8º), all the data were brought to a common 2.8º x 2.8º spatial grid using bilinear interpolation prior to processing. Moreover, as the minimum time period covered by the simulations is 30 years, for all simulations the first 30 years were selected for consistency. Results from the ensemble of all the models are presented within the manuscript on an annual and seasonal basis (winter vs summer) while

results for each model separately are given in the supplement. The statistical significance of the results at the 95% confidence level is checked using a paired sample two-sided t-test.

To decompose the effect of different present-day aerosol types on early industrial climate, supplementary data from other 3 experiments, namely piClim-SO$_2$ (all forcings set to 1850 but using SO$_2$ precursor emissions of the year 2014), piClim-BC (all forcings set to 1850 using BC precursor emissions of the year 2014) and piClim-OC (all forcings set to 1850 using OC

precursor emissions of the year 2014), were used. At the time this manuscript was written, there were available data from 3 models only (CNRM-ESM2-1, MRI-ESM2-0 and NorESM2-LM). Similarly, by subtracting the piClim-control simulation from these 3 simulations, the fast responses of pre-industrial climate to present-day sulfates, BC and OC aerosols are calculated. Supporting information for each model, the corresponding experiments and the model basic references/dois are shown in Table 1.

Taking into consideration that the perturbation experiments to the control simulation are based on emissions for aerosol precursors set to present-day (2014), Figure 1 shows the annual SO$_2$ and BC emissions for 2014 used in piClim-aer simulations as well as the differences from their respective emissions for year 1850 used in piClim-control simulations. Figure 1 is based on the emissions used in CNRM-ESM2-1, but the emissions are similar for the rest of the models used here, indicating that the largest present-day sources of SO$_2$ are over East Asia, India, North America and Europe, while for BC over





East Asia, India and Africa. The differences between piClim-aer and piClim-control in SO$_2$ emissions are peaking over East Asia, India, North America and Europe, while for BC the emissions peak over East Asia, India and spot regions in Africa. The differences between piClim-aer and piClim-control in BC emissions are very low over Europe and North America.

# 3 Results and discussion

## 3.1 Changes in net radiative flux at TOA

The difference between piClim-aer and piClim-control simulations in the TOA radiative flux including both the shortwave (SW) and longwave (LW) was calculated for each one of the models to estimate the total aerosol ERF following Forster et al. (2016). The ensemble mean of the aerosol ERF from the 10 models is shown in Figure 2 on an annual basis as well as for the boreal winter/austral summer period including the months December, January, February (DJF) and for the boreal summer/austral winter period including the months June, July, August (JJA). The mean ERF values (global, Northern Hemisphere and South Hemisphere) for each model on an annual basis, DJF and JJA are shown in Table 2. Overall, on an annual basis (Figure 2a), we see a characteristic spatially extensive negative ERF at the TOA over the globe induced by the perturbation of the present day aerosols (global annual average ERF of -1 W m$^{-2}$), especially over the Northern Hemisphere (NH annual average ERF of -1.5 W m$^{-2}$) with the largest negative ERF values over East Asia in response to the SO$_2$ emissions. The global annual average of all aerosols ERF (-1 W m$^{-2}$) is similar to the multi-model mean value of 8 ACCMIP models in IPCC AR5 (-1.17 W m$^{-2}$) with the patterns being also similar (Shindell et al., 2013).The negative ERF values over the Northern Hemisphere generally become stronger during the boreal summer with regional maxima over East Asia and India (Figure 2c). The negative values of ERF persist over East Asia during DJF (Figure 2b). Figure 2a also shows a characteristic positive ERF over reflective continental surfaces such as the Sahara Desert, Greenland and Alaska. This positive ERF over reflective continental surfaces of the NH becomes also stronger during JJA when the levels of radiation peak (Figure 2c). The positive ERF values over the reflective continental surfaces can be explained by the fact that the very high surface albedo reduces the effect of scattering aerosols, while increasing the effect of absorbing aerosols, leading to a net positive forcing.

The aerosol perturbation ERFs on an annual basis for each one of the models used in the ensemble are illustrated in Figure 3. Figure S1 and Figure S2 of the supplementary material show the aerosol perturbation ERFs for each model, for DJF and for JJA, respectively. Despite regional differences the spatially extensive negative ERF at the TOA over continental areas with the largest negative ERF values over East Asia and the positive ERF over the Sahara Desert are robust features for all models on an annual basis (Figure 3) and JJA (Figure S2). Positive ERF values over the Saudi Arabia Desert, Greenland and Alaska are also seen in the majority of models on an annual basis (Figure 3) with this signal becoming more robust and stronger during JJA (Figure S2). In DJF there are also common features among the models as the negative ERF values over East Asia, southern Africa and South America (Figure S1).



Figure 4 provides a comprehensive multi-model outlook of the zonal mean aerosol TOA ERF (with ±1σ range of the 10 models), with the largest negative values found over the mid-latitudes of the Northern Hemisphere (40º N) for the annual analysis (-2.1 W m$^{-2}$ at 40º N), for JJA (-2.7 W m$^{-2}$ at 40º N) and for DJF (-1.3 W m$^{-2}$ at 30º N).

In this study, the piClim-SO$_2$, piClim-BC and piClim-OC simulations were not available for all the participating models to decompose their respective ERF responses. Nevertheless, the available piClim-SO$_2$, piClim-BC and piClim-OC simulations for CNRM-ESM2-1, MRI-ESM2-0 and NorESM2-LM (Figure S9) show that their sulfate ERF patterns are similar to the all-aerosol ERF patterns (Figure 3) indicating the dominating role of sulfates in the all-aerosols ERF.

## 3.2 Near surface temperature changes

The fast temperature responses on pre-industrial climate due to present day aerosols are illustrated in Figure 5 with the differences between piClim-aer and piClim-control in near surface temperature for the ensemble of the 10 models on an annual basis as well as for DJF and JJA, separately. The mean fast temperature response values (global, Northern Hemisphere and South Hemisphere) for each model on an annual basis, DJF and JJA are shown in Table 2. On an annual basis (Figure 5a) there is an overall cooling over the continental areas especially in the Northern Hemisphere with the largest cooling over East Asia and India in response to the SO$_2$ emissions and the pattern of ERF. The cooling in the Northern Hemisphere is generally enhanced during the boreal summer (Figure 5c) following the more negative ERF values presented in section 3.1 for this season (Figure 2c). The zonal means of the fast temperature responses (Figure 4) reveal a general cooling over the mid-latitudes in the Northern Hemisphere on an annual basis (up to -0.12 ºC at 45º N), during JJA (-0.2 ºC at 45º N) and during DJF (-0.1 ºC at 30º N). These values are lower compared to other multi-model studies that incorporate both fast and slow responses. For example, multi-model sensitivity experiments with perturbations in anthropogenic emissions of SO$_2$, BC and OC showed that the removal of present-day anthropogenic aerosol emissions induces a global mean surface heating of 0.5-1.1°C, with sulfate aerosols being the dominant surface air temperature driver for the present-day emissions (Samset et al., 2018). Another multi-model study indicated a global mean surface temperature increase of 0.7 °C in response to the reduction in SO$_2$, with the zonal mean temperature changes increasing with latitude up to a value of around 2.5 °C at the North Pole (Baker et al., 2015). In a recent modelling study it was shown that removing SO$_2$ emissions from any of the main emission regions in the northern-hemisphere (North America, Europe, East and South Asia) results in significant warming across the northern hemisphere with a preferred spatial pattern, yet a varying magnitude (Kasoar et al., 2018). Simulated surface temperature changes due to the removal of U.S. anthropogenic SO$_2$ emissions revealed robust patterns of temperature responses over land, with increases in temperature for most of the Northern Hemisphere land regions and the strongest response towards the Arctic (Conley et al., 2018; Shindell et al., 2015).

The fast temperature responses for each one of the models on annual basis are illustrated in Figure 6, while Figure S3 and Figure S4 of the supplementary material show the respective patterns for, DJF and JJA. Most models show continental cooling on an annual basis with a robust feature of cooling over East Asia (Figure 6). However, there are regional



differences among the models in the pattern of induced fast temperature responses over Europe, North America and Africa (Figure 6). The continental cooling in the Northern Hemisphere becomes stronger in JJA but still we note regional differences in the pattern of fast temperature responses over Europe, North America and Africa (Figure S4). The available piClim-$SO_2$, piClim-BC and piClim-OC simulations of 3 models (CNRM-ESM2-1, MRI-ESM2-0 and NorESM2) show that

the patterns of temperature differences between piClim-$SO_2$ and piClim-control resemble the patterns of the differences between piClim-aer and piClim-control (Figure S10 versus Figure 6). This is in line with previous multi-model studies showing that sulfates are the dominant aerosol surface temperature driver for the present-day emissions (Baker et al., 2015; Samset et al., 2018).

It is interesting to note the slight warming seen in the Arctic on the annual basis (Figure 5a) which is not apparent in

JJA (Figure 5c) but becomes stronger in DJF (Figure 5b). Practically, the DJF warming signal determines the annual warming signal over the Arctic. As can be seen in Figure 4, in the northern polar latitudes there is a warming signal in the annual (up to 0.25 ℃) and in DJF (up to 0.45 ℃) but the model range is large. The pattern of ERF perturbation over the Arctic in DJF (Figure 2b) cannot explain this warming signal, but the aerosol induced circulation changes discussed in section 3.3 provide a plausible explanation. Specifically, the wind vector and geopotential height (GH) differences at 850

hPa between piClim-aer and piClim-control (Figure 9b) reveal a positive GH anomaly (anticyclonic anomaly) over northern Siberia and part of the Arctic which could induce adiabatic heating of the subsiding air. Furthermore, there is a characteristic dipole pattern with intensification of the Icelandic Low (cyclonic anomaly) and an anticyclonic anomaly over Southeastern Europe inducing warm air advection towards the northern polar latitudes. Although sea-ice is fixed in these simulations, snow and ice over land can change from the Arctic warming, thus activating albedo feedbacks which could further amplify

the warming signal.

Several models show this slight warming in regions of the Arctic on an annual basis, with CESM2 and NorESM2 revealing the largest warming signal (Figure 6). This feature is stronger and more robust among the models during DJF (Figure S3) implying the role of circulation changes rather than ERF as a plausible cause. For example, for CESM2 and NorESM2 there is no positive ERF to account for the Arctic warming (Figure S1) but the DJF circulation anomalies (Figure

S7) reveal a cyclonic (lower GH) anomaly over Europe which in association with an anticyclonic anomaly over Siberia induces a warm advection at the eastern side of the cyclonic anomaly (or western side of the anticyclonic anomaly) towards the polar regions. Furthermore, the available piClim-$SO_2$, piClim-BC and piClim-OC simulations for NorESM2 (Figure S10) show that the pattern of Arctic warming seen from the temperature differences between piClim-aer and piClim-control (Figure 6) is similar to the pattern of temperature differences between piClim-$SO_2$ and piClim-control and not to either

piClim-BC or piClim-OC (Figure S10). So, the perturbation experiment with present-day BC emissions cannot justify this warming in NorESM2. This warming signal in the Arctic in response to present-day cooling aerosols is also seen in a PDRMIP multi-model study from the pattern of fast temperature responses (with fixed SST) in perturbation experiments with a five-fold increase in $SO_4$ over Asia or Europe (see Figure 2 in Liu et al., 2018). However, this is not a robust result as it is not evident in other previous multi-model perturbation experiments. For example, in a recent multi-model study for the



Arctic, perturbation experiments with fixed SSTs applying a 10-fold increase in BC concentrations/emissions and a 5-fold increase in $SO_4$ concentrations/emissions showed a temperature increase of roughly 0.2 ℃ and a temperature decrease of roughly -0.3 ℃, respectively (see Figure S2 in Stjern et al., 2019). Stjern et al., (2019) noted the large inter-model range in both slow and fast temperature responses over Arctic, showing that the fast temperature responses are very small compared to the slow responses. Model sensitivity experiments by increasing $SO_2$ in Europe, North America, East Asia and South Asia showed a consistent cooling almost everywhere in the Northern Hemisphere, with the Arctic exhibiting the largest temperature response in all experiments but these results were considering both slow and fast temperature responses (Lewinschal et al., 2019). There are also single-model studies (Sand et al., 2013) and multi-model studies (Stjern et al., 2017) indicating relatively large responses in the Arctic to BC perturbation, but with particularly large inter-model range.

## 3.3 Precipitation and circulation changes

The fast precipitation responses on pre-industrial climate due to present day aerosols are illustrated in Figure 7 with the differences between piClim-aer and piClim-control in precipitation for the ensemble of the 10 models on an annual basis as well as for DJF and JJA. Similarly, Figure 9 shows the respective fast circulation responses based on aerosol induced changes in the wind vectors and GH at 850 hPa. Generally, the largest fast precipitation responses are seen in the tropical belt regions with the highest precipitation rates, while the shift in pattern of these responses from DJF to JJA is linked with the northward movement of ITCZ from winter to summer (Figure 7). The mean fast precipitation response values (global, Northern Hemisphere and South Hemisphere) for each model on an annual basis, DJF and JJA are shown in Table 2.

These fast precipitation responses are characterized by a reduction over the continental regions (Figure 7) in agreement with previous studies (Westervelt et al., 2018). The pattern of annual precipitation responses (Figure 7a) is very similar to the pattern of the fast precipitation experiments with a five-fold increase in $SO_4$ in a PDRMIP multi-model study (see Figure 4 from Samset et al., 2016). The zonal means of fast precipitation responses on an annual basis show overall small reductions over the Northern Hemisphere, but the key feature is the appearance of the larger changes in the tropical belt with a southward shift of a decrease-increase pattern (Figure 4a). This pattern exhibits similarities with Figure 2d from Hwang et al. (2013) indicating that anthropogenic aerosol cooling of the Northern Hemisphere is the primary cause of a consistent southward shift of the tropical rain belt across GCMs. The dimming over the Northern Hemisphere causes a relative cooling of the Northern Hemisphere compared to the Southern Hemisphere, which induces a southward shift of the northern edge of the tropical rain belt. Myhre et al. (2016) noted that over land, increased anthropogenic sulfate aerosols induce generally reduced precipitation, such as over equatorial Africa or South Asia. Multi-model studies show that the southward/northward shift of ITCZ in the sulfate increase/decrease perturbation experiments is a robust feature among many models (Liu et al., 2018b; Chen et al., 2018) in response to hemispherically asymmetric cooling from a Northern Hemisphere aerosol perturbation (e.g., Acosta Navarro et al. 2017; Allen et al. 2015), leading the ITCZ and tropical precipitation to shift away from the cooled hemisphere patterns (Rotstayn et al., 2015; Undorf et al., 2018; Westervelt et al., 2018). In another





mutli-model study, it was shown that in response to an idealized anthropogenic aerosol, fast and slow ITCZ shifts oppose each other with the slow ITCZ southward shift dominating over the small fast northward ITCZ shift (Voigt et al., 2017). The small fast ITCZ northward shift differs from our results but in the study by Voigt et al. (2017) only aerosol-radiation interactions were considered. Allen and Ajoku (2016) reported that the increase in aerosols over the twentieth century has

led to contraction of the northern tropical belt, thereby offsetting part of the widening associated with the increase in GHGs. These processes partially also explain the southward shift of the NH tropical edge from the 1950s to the 1980s (Allen et al., 2014; Brönnimann et al., 2015) and the severe drought in the Sahel that peaked in the mid-1980s (Rotstayn and Lohmann, 2002; Undorf et al., 2018b).

On an annual basis it is characteristic that the dipole pattern of precipitation decreases over East Asia and increases

over southern India, Bay of Bengal and South China Sea (Figure 7a). This signal gets stronger during JJA in the monsoon season (Figure 7c). The zonal mean precipitation changes in JJA (Figure 4c) show a shift from -0.13 mm/day at 30° N to 0.04 mm/day at 15° N, which can be justified by the dipole pattern of precipitation decrease over East Asia and increase over southern India, Bay of Bengal and South China Sea in summer season. This pattern of precipitation decreases over East Asia and increases over southern India, the Bay of Bengal and South China Sea is rather a robust feature in all model simulations

for the annual basis (Figure 8) and the monsoon season (Figure S6). This dipole pattern of JJA precipitation responses over East Asia (Figure 7c) is similar to the pattern of fast precipitation responses and of the changes in column-integrated dry static energy flux divergence over this region in perturbation experiments with a five-fold increase in $SO_4$ over Asia in a PDRMIP multi-model study (see Figure1 and Figure 7 from Liu et al., 2018).

Figure 9a (annual basis) indicates a positive GH anomaly (anticyclonic anomaly) over East Asia and north India with the

horizontal wind vector anomalies implying weakening of the monsoon winds. This is also supported by the weakening of the upward motions over East Asian, as derived from the piClim-aer and piClim-control differences in vertical velocities (not shown).

The above-mentioned signals for the annual basis become stronger during the monsoon season JJA (Figure 9c) with the GH and wind vectors anomalies implying a southward shift of the ITCZ and a weakening of the Indian and East Asian

monsoon systems. It should be also noted that this is a rather robust feature for all models in JJA (Figure S8), verifying that the overall negative radiative forcing of aerosols over Northern Hemisphere shifts southward the ITCZ and weakens the Asian monsoon systems. This is a characteristic and robust feature among most of the models utilized in our study, which is associated with a southward shift of the ITCZ and weakening of Indian and East Asian monsoon systems, verifying that the overall negative radiative forcing of aerosols over the Northern Hemisphere shifts southward the ITCZ and weakens the

Asian monsoon systems. Summer monsoons rainfall is caused by the faster solar heating of subtropical land compared to the adjacent oceans, which causes convergence and rising of the moist marine air over land. Therefore, the dimming weakens the monsoon flow and precipitation. This has been noted in several previous studies. The response to Asian and European $SO_2$ emissions leads to cooling of East Asia and a weakening of the East Asian summer monsoon with decrease of precipitation over East Asia, and an increase to the south and over the Western North Pacific (Dong et al., 2016). Bartlett et





al., (2018) also show that increased sulfate aerosols in SO₂ emission sensitivity experiments under RCP2.6 will lead to surface cooling and weakening of the East Asian monsoon circulation. These processes explain the observed decrease of southeast Asian Monsoon precipitation during the second half of the 20th century (Bollasina et al., 2011; Krishnan et al., 2016; Lau and Kim, 2017; Lin et al., 2018; Sanap, 2015; Takahashi et al., 2018; Undorf et al., 2018b).

5        Another feature in the fast precipitation responses is the relative drying over Africa (southward of Sahel) on an annual basis (Figure 7a), which is apparently a robust signal in all models (Figure 8). The ensemble drying signal shifts from Sahel in boreal summer JJA (Figure 7c) to southern Africa in austral summer DJF (Figure 7b). Most of the models show also this southward drying shift from JJA (Figure S6) to DJF (Figure S5). Specifically, the slight Sahel drying in JJA (Figure 7c) is associated with Sahel cooling (Figure 5c), with positive GH anomalies (Figure 9c) and in terms of circulation changes

with a slight weakening of the easterlies and of the West African monsoon (Figure 6c). The drying in southern Africa during austral summer DJF (Figure 7b) is associated with positive GH anomalies and with a weakening of the Southeast African monsoon winds (Figure 9b). These results agree with studies showing a weakening of the West African monsoon and a decrease in the Sahel precipitation with increasing SO₂ emissions (Dong et al., 2014). In response to U.S. SO₂ emission reductions (opposite to the perturbation in our study) a northward shift of the tropical rain belt and the ITCZ was also noted

delivering additional wet season rainfall to the Sahel (Westervelt et al., 2017). Similarly, other recent studies showed that in the West Africa and the Sahel, precipitation may increase in response to the aerosol reductions in remote regions, because an anomalous interhemispheric temperature gradient alters the position of the ITCZ (Undorf et al., 2018; Westervelt et al., 2018).

         Figure 7a shows a relative drying over Central and South America on an annual basis with the drying seen more

clearly over Central America in boreal summer JJA (Figure 7c), and over South America in austral summer DJF (Figure 7b). Despite the regional differences, these features are common in most of the models in JJA (Figure S6) and DJF (Figure S5). The Central America drying in boreal summer JJA (Figure 7c) is associated with a positive GH anomaly and weakening of the dominating westerly flow (Figure 9c), as well as weakening of the upward motions (not shown). These changes imply a weakening of the North American Monsoon. The South America drying in austral summer DJF (Figure 7b) is associated

with a relevant cooling (Figure 5b), and in terms of circulation changes with a weakening of the dominating westerly flow and of the South American Monsoon winds (Figure 9b).

         An interesting feature in the wind vector and geopotential height (GH) differences at 850 hPa between piClim-aer and piClim-control is the anticyclonic anomaly over northern Siberia and part of the Arctic and a characteristic dipole pattern, with intensification of Icelandic Low (cyclonic anomaly) and an anticyclonic anomaly over Southeastern Europe,

inducing warm air advection towards the northern polar latitudes in DJF (Figure 9b). The intensification of the Icelandic Low (cyclonic anomaly) is also apparent in the annual basis (Figure 9a) and in JJA (Figure 9c) and can be noted in the majority of models (although with spatial shifts) for the annual analysis (Figure 10), for DJF (Figure S7) and for JJA (Figure S8). The pattern of aerosol induced circulation changes in DJF looks similar with the pattern of geopotential height changes at 500 hPa in simulations with predominantly scattering aerosols and opposite in simulations with predominantly absorbing





aerosols (see Figure 12 in Allen and Sherwood, 2011). The deepening of the Icelandic Low is also apparent in the the pattern of changes in sea level pressure in perturbation experiments with a five-fold increase in SO$_4$ over Asia or Europe in a PDRMIP multi-model study (see Figure 7 from Liu et al., 2018). Furthermore, the circulation changes in piClim-aer simulations with the intensification of the Icelandic Low are in line with aerosol induced circulation changes in CMIP6 hist simulations (with opposite sign) in response to aerosol reductions from 1990 to 2020 (Allen et al., in preparation 2019). Hence, it appears that CMIP6 simulations are suggesting a deepening of the Icelandic Low in response to an increase in aerosols and a weakening in response to aerosol decreases. Figure 9b indicates also a slight weakening of the Aleutian Low in DJF but this is not a robust feature for all models (Figure S7).

## 4. Conclusions

In this work, we use the CMIP6 simulations from 10 different models (CanESM5, CESM2, CNRM-CM6-1, CNRM-ESM2-1, GISS-E2-1-G, IPSL-CM6A-LR, MIROC6, MRI-ESM2-0, NorESM2-LM, UKESM1-0-LL) to study the fast responses on pre-industrial climate due to the present day aerosols. All models carried out two sets of simulations; the piClim-control (with all forcings set to the year 1850 using aerosol precursor emissions of 1850) and the piClim-aer (again with all forcings set to 1850 but using aerosol precursor emissions of the year 2014).

The perturbation by the present-day aerosols indicates negative TOA ERF values around the globe, especially over continental regions of the Northern Hemisphere in summer, with the largest negative values over East Asia in response to the SO$_2$ emissions. Simulations in 3 models (CNRM-ESM2-1, MRI-ESM2-0 and NorESM2-LM) with individual perturbation experiments using present day SO$_2$, BC and OC emissions show the dominating role of sulfates in all-aerosols ERF.

In response to the pattern of all-aerosol ERF, the fast temperature responses are characterised by cooling over the continental areas, especially in the Northern Hemisphere with the largest cooling over East Asia and India. The zonal means of the fast temperature responses reveal a general cooling over the mid-latitudes in the Northern Hemisphere up to -0.12 ºC at 45º N on an annual basis, up to -0.2 ºC at 45º N during boreal summer and up to -0.1 ºC at 30º N during boreal winter. The available piClim-SO$_2$, piClim-BC and piClim-OC simulations of 3 models show that sulfates are the dominant aerosol surface temperature driver for the present-day emissions.

In the northern polar latitudes, there is a warming signal on an annual basis (up to 0.25 ºC) and for DJF (up to 0.45 ºC) but the model range is large. This Arctic warming signal in DJF is not justified by the regional ERF signal but is presumably linked to aerosol induced circulation changes causing adiabatic heating of the subsiding air over northern Siberia and part of the Arctic, as well as warm air advection from Europe towards the northern polar latitudes. NorESM2 is one of the models showing a strong warming in the Arctic, but the perturbation experiment with present day BC emissions cannot justify this warming. Instead, the pattern of Arctic warming seen from the temperature differences between piClim-aer and piClim-control is resembled by the perturbation experiment with present day SO$_2$ emissions.



The largest fast precipitation responses are seen in the tropical belt regions, generally characterized by reduction over the continental regions. The zonal mean of fast precipitation responses on an annual basis shows overall small reductions over the Northern Hemisphere, but the characteristic feature is the appearance of the larger changes in the tropical belt with a dipole decrease-increase pattern in response to a southward shift of the ITCZ.

The zonal mean precipitation changes in boreal summer show a shift from -0.13 mm/day at 30º N to 0.04 mm/day at 15º N, which can be largely justified by the dipole pattern of precipitation decrease over East Asia and increase over southern India, the Bay of Bengal and South China Sea. This is a characteristic and robust feature among most of the models utilized in this study, verifying that the overall negative radiative forcing of aerosols over the Northern Hemisphere shifts the ITCZ southward and weakens the Asian monsoon systems. Summer monsoons rainfall is caused by the faster solar heating of subtropical land compared to the adjacent oceans, which causes convergence and rising of the moist marine air over land. Therefore, the dimming weakens the monsoon flow and precipitation.

It is also noticed that most models in this study yield a drying signal in Africa, shifting from Sahel in boreal summer JJA to southern Africa in austral summer DJF, linked to a weakening of the West African and Southeast African monsoon systems, respectively. Furthermore, we note a drying signal in America, shifting from Central America in boreal summer JJA to South America in austral summer DJF, which is also associated with circulation changes inducing a weakening of the North American and South American Monsoon winds.

An interesting feature in aerosol induced circulation changes is the characteristic dipole pattern with intensification of the Icelandic Low (cyclonic anomaly) and an anticyclonic anomaly over Southeastern Europe, inducing warm air advection towards the northern polar latitudes in DJF. It appears that the deepening of the Icelandic Low in response to an increase in aerosols is a robust feature in the simulations.

## Acknowledgements

This research was funded by the project "PANhellenic infrastructure for Atmospheric Composition and climatE change" (MIS 5021516) which is implemented under the Action "Reinforcement of the Research and Innovation Infrastructure", funded by the Operational Programme "Competitiveness, Entrepreneurship and Innovation" (NSRF 2014-2020) and co-financed by Greece and the European Union (European Regional Development Fund). D. O. and M. S. were supported by the Research Council of Norway (grant nos. 229771, 285003, and 285013), by Notur/NorStore (NN2345K and NS2345K), and through EU H2020 grant no. 280060.



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



**Table 1**: Information on models resolution, vertical levels, model simulations and references. Each experiment has a variant label resembled by r<k>i<l>p<m>f<n> where k = realization_index, l = initialization_index, m = physics_index and n = forcing_index.

| Model | Resolution | Vertical levels | Model type | piClim-control Variant label | piClim-aer Variant label | piClim-SO₂ Variant label | piClim-BC Variant Label | piClim-OC Variant label | Reference/doi |
|---|---|---|---|---|---|---|---|---|---|
| CanESM5 | 2.8º x 2.8º; | 49 levels; top level 1 hPa | ESM interactive chemistry | r1i1p2f1 | r1i1p2f1 | | | | Cole et al., 2019a,b |
| CESM2 | 0.95º x 1.25º | 32 levels; top level 2.25 hPa | ESM interactive aerosols | r1i1p1f1 | r1i1p1f1 | | | | CESM2, 2018a,b |
| CNRM-CM6-1 | 1.4º x 1.4º | 91 levels; top level 78.4 km | GCM no interactive aerosols | r1i1p1f2 | r1i1p1f2 | | | | Voldoire, 2019a,b |
| CNRM-ESM2-1 | 1.4º x 1.4º | 91 levels; top level 78.4 km | ESM fully interactive aerosols | r1i1p1f2 | r1i1p1f2 | r1i1p1f2 | r1i1p1f2 | r1i1p1f2 | Seferian, 2019a,b Seferian et al., 2019 Michou et al., 2019 |
| GISS-E2-1-G | 2º x 2.5º | 40 levels; top level 0.1 hPa | GCM no interactive aerosols | r1i1p1f1 | r1i1p1f1 | | | | GISS, 2019a,b Kelley et al., 2020 Bauer and Tsigaridis, 2020 |
| IPSL-CM6A-LR | 1.27º x 2.5º | 79 levels; top level 80 km | GCM prescribed aerosols | r1i1p1f1 | r1i1p1f1 | | | | Boucher et al., 2018 Boucher et al., 2019 |
| MIROC6 | 1.4º x 1.4º | 81 levels; top level 0.004 hPa | GCM interactive aerosols | r1i1p1f1 | r1i1p1f1 | | | | Sekiguchi and Hideo, 2019a,b |
| MRI-ESM2-0 | 1.125º x 1.125º | 80 levels; top level 0.01 hPa | ESM interactive aerosols | r1i1p1f1 | r1i1p1f1 | r1i1p1f1 | r1i1p1f1 | r1i1p1f1 | Yukimoto et al., 2019a,b |
| NorESM2-LM | 1.9º x 2.5º | 32 levels; top level 3 hPa | ESM interactive aerosols | r1i1p1f1 | r1i1p1f1 | r1i1p1f1 | r1i1p1f1 | r1i1p1f1 | NorESM2-LM, 2018a,b Kirkevåg et al., 2018 |
| UKESM1-0-LL | 1.25º x 1.875º | 85 levels; top level 85 km | ESM interactive aerosols | r1i1p1f2 | r1i1p1f2 | | | | O'Connor, 2019a,b |





Table 2: Differences between piClim-aer and piClim-control in temperature, precipitation and ERF for each model on an annual basis, DJF and JJA. The values are given as mean values for global, northern hemisphere (NH) and southern hemisphere (SH).

| annual | Temperature (ºC) | | | Precipitation (mm/day) | | | ERF (W m⁻²) | | |
|---|---|---|---|---|---|---|---|---|---|
| | GLOBAL | NH | SH | GLOBAL | NH | SH | GLOBAL | NH | SH |
| CanESM5 | -0.01 | -0.02 | 0 | -0.03 | -0.04 | -0.02 | -0.84 | -1.22 | -0.47 |
| CESM2 | 0.04 | 0.08 | -0.01 | -0.03 | -0.03 | -0.03 | -0.8 | -0.82 | -0.79 |
| CNRM-CM6-1 | -0.06 | -0.11 | 0 | -0.01 | -0.01 | -0.01 | -1.15 | -1.59 | -0.72 |
| CNRM-ESM2-1 | -0.05 | -0.08 | -0.01 | -0.01 | -0.01 | -0.01 | -0.74 | -1.04 | -0.44 |
| GISS-E2-1-G | -0.06 | -0.11 | -0.02 | 0 | 0.01 | -0.01 | -1.31 | -1.97 | -0.64 |
| IPSL-CM6A-LR | -0.04 | -0.08 | -0.01 | -0.01 | -0.02 | 0 | -0.59 | -0.95 | -0.23 |
| MIROC6 | -0.05 | -0.1 | 0 | -0.01 | -0.01 | -0.01 | -1.06 | -1.54 | -0.58 |
| MRI-ESM2-0 | -0.02 | -0.06 | 0.02 | -0.03 | -0.04 | -0.02 | -1.18 | -1.91 | -0.45 |
| NorESM2-LM | 0.01 | 0.02 | 0 | -0.01 | -0.02 | 0 | -1.2 | -2.03 | -0.38 |
| UKESM1-0-LL | -0.04 | -0.07 | -0.02 | -0.02 | -0.01 | -0.02 | -1.11 | -1.5 | -0.72 |
| **ENSEMBLE** | **-0.03** **±0.03** | **-0.05** **±0.06** | **-0.01** **±0.01** | **-0.02** **±0.01** | **-0.02** **±0.02** | **-0.01** **±0.01** | **-1.00** **±0.24** | **-1.46** **±0.44** | **-0.54** **±0.18** |
| DJF | Temperature | | | Precipitation | | | ERF | | |
| | GLOBAL | NH | SH | GLOBAL | NH | SH | GLOBAL | NH | SH |
| CanESM5 | 0 | -0.01 | 0 | -0.02 | -0.03 | -0.01 | -0.5 | -0.56 | -0.44 |
| CESM2 | 0.09 | 0.21 | -0.03 | -0.02 | -0.02 | -0.03 | -0.52 | -0.5 | -0.55 |
| CNRM-CM6-1 | -0.06 | -0.09 | -0.02 | -0.01 | -0.01 | 0 | -1.05 | -1.38 | -0.71 |
| CNRM-ESM2-1 | -0.05 | -0.1 | 0 | -0.02 | 0 | -0.03 | -0.56 | -0.72 | -0.39 |
| GISS-E2-1-G | -0.02 | -0.03 | -0.01 | 0.01 | 0 | 0.01 | -1.11 | -1.42 | -0.81 |
| IPSL-CM6A-LR | -0.06 | -0.08 | -0.04 | -0.01 | -0.01 | 0 | -0.4 | -0.56 | -0.24 |
| MIROC6 | -0.01 | -0.03 | 0 | -0.01 | -0.01 | 0 | -0.95 | -1.08 | -0.81 |
| MRI-ESM2-0 | -0.03 | -0.06 | 0.01 | -0.03 | -0.03 | -0.03 | -0.74 | -1.09 | -0.39 |
| NorESM2-LM | 0.07 | 0.12 | 0.01 | 0 | -0.01 | 0.01 | -1.02 | -1.72 | -0.32 |
| UKESM1-0-LL | -0.02 | 0 | -0.03 | -0.01 | 0 | -0.02 | -0.76 | -0.88 | -0.64 |
| **ENSEMBLE** | **-0.01** **±0.05** | **-0.01** **±0.10** | **-0.01** **±0.02** | **-0.01** **±0.01** | **-0.01** **±0.01** | **-0.01** **±0.02** | **-0.76** **±0.26** | **-0.99** **±0.42** | **-0.53** **±0.20** |
| JJA | Temperature | | | Precipitation | | | ERF | | |
| | GLOBAL | NH | SH | GLOBAL | NH | SH | GLOBAL | NH | SH |
| CanESM5 | -0.02 | -0.05 | 0.01 | -0.04 | -0.05 | -0.02 | -1.35 | -2.26 | -0.44 |
| CESM2 | 0 | -0.01 | 0.01 | -0.04 | -0.06 | -0.01 | -0.55 | -0.49 | -0.62 |
| CNRM-CM6-1 | -0.06 | -0.12 | -0.01 | -0.01 | -0.01 | -0.01 | -1.2 | -1.81 | -0.59 |
| CNRM-ESM2-1 | -0.06 | -0.08 | -0.03 | -0.01 | -0.02 | -0.01 | -0.76 | -1.17 | -0.34 |
| GISS-E2-1-G | -0.09 | -0.13 | -0.04 | 0 | 0.01 | -0.01 | -1.21 | -2.09 | -0.32 |
| IPSL-CM6A-LR | -0.04 | -0.09 | 0.01 | -0.01 | -0.03 | 0.01 | -0.86 | -1.35 | -0.36 |





| | | | | | | | | | |
|---|---|---|---|---|---|---|---|---|---|
| MIROC6 | -0.09 | -0.18 | -0.01 | -0.01 | -0.01 | -0.01 | -0.93 | -1.61 | -0.26 |
| MRI-ESM2-0 | -0.01 | -0.06 | 0.04 | -0.03 | -0.06 | 0 | -1.65 | -2.8 | -0.51 |
| NorESM2-LM | -0.05 | -0.09 | 0 | -0.02 | -0.05 | 0.01 | -1.22 | -2.01 | -0.42 |
| UKESM1-0-LL | -0.08 | -0.12 | -0.03 | -0.02 | -0.03 | -0.02 | -1.54 | -2.42 | -0.67 |
| **ENSEMBLE** | **-0.05** **±0.03** | **-0.09** **±0.05** | **-0.01** **±0.02** | **-0.02** **±0.01** | **-0.03** **±0.02** | **-0.01** **±0.01** | **-1.13** **±0.35** | **-1.80** **±0.67** | **-0.45** **±0.14** |





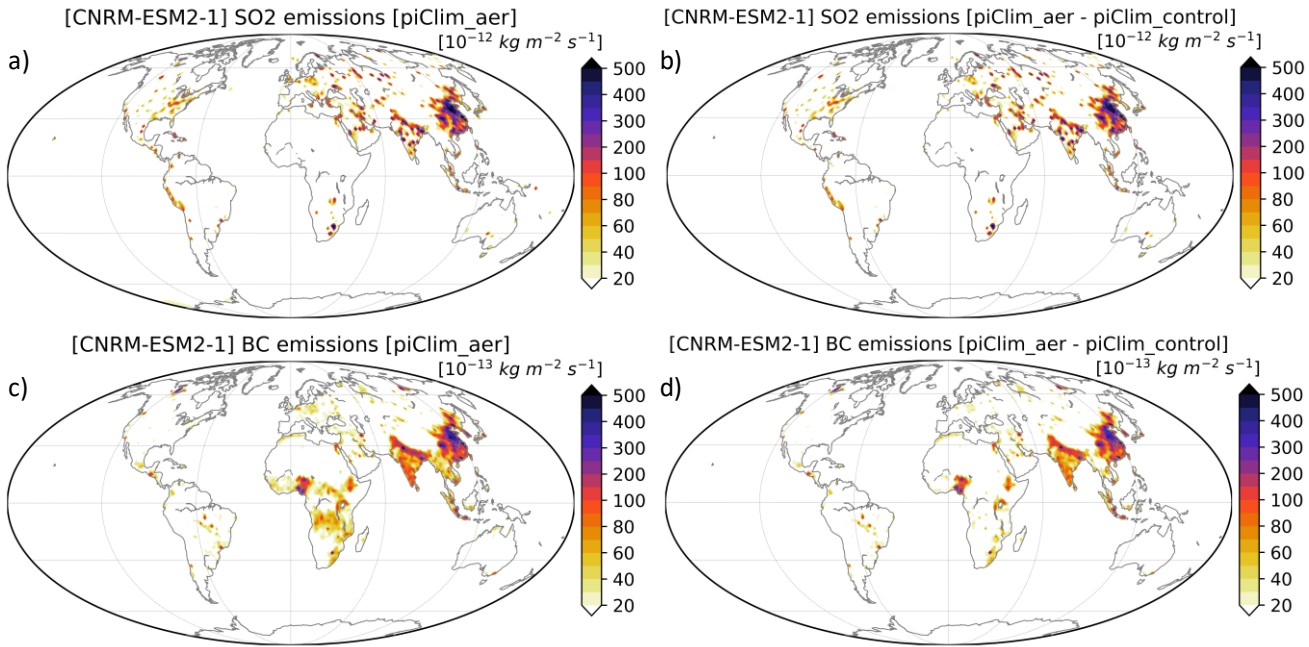

**Figure 1:** Annual $SO_2$ (in $10^{-12}$ Kg m$^{-2}$ s$^{-1}$) and BC emissions (in $10^{-13}$ Kg m$^{-2}$ s$^{-1}$) for 2014 (a and c) used in CNRM-ESM2-1 piClim-aer simulation and differences in annual $SO_2$ and BC emissions between year 2014 (in piClim-aer) and year 1850 (in piClim-control) (b and d). Mind that the scale for BC emissions is by a factor of 10 lower than for $SO_2$ emissions.





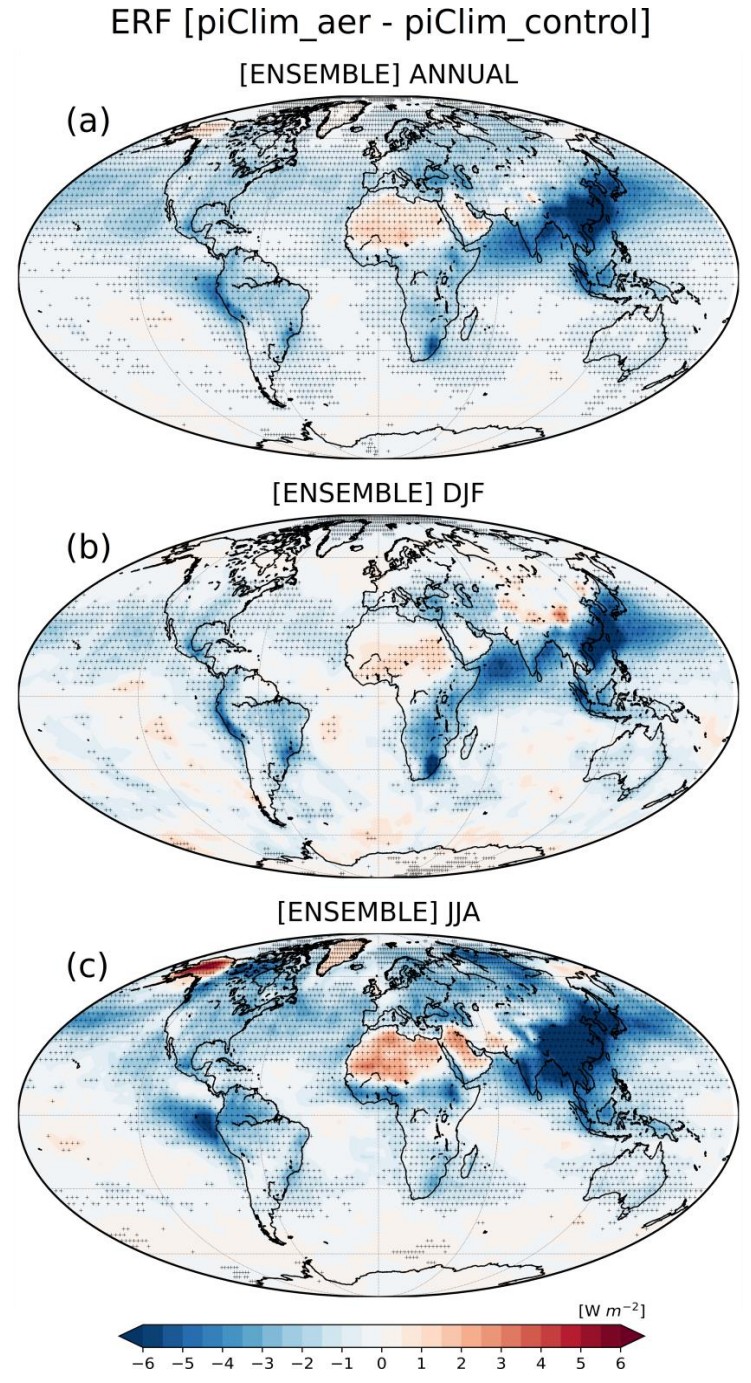

**Figure 2:** Differences between piClim-aer and piClim-control in the net radiative flux (W m-2) at TOA including both SW and LW (all-aerosol ERF) for the ensemble of 10 models on an annual basis (a). for DJF (b) and for JJA (c). The dot shading indicates areas in which the differences are statistically significant at the 95% confidence level.





## ERF ANNUAL [piClim_aer - piClim_control]

**Figure 3.** Annual differences between piClim-aer and piClim-control in the net radiative flux (W m⁻²) at TOA including both SW and LW (aerosol ERF) for each one of the models used for the ensemble. The dot shading indicates areas in which the differences are statistically significant at the 95% confidence level.

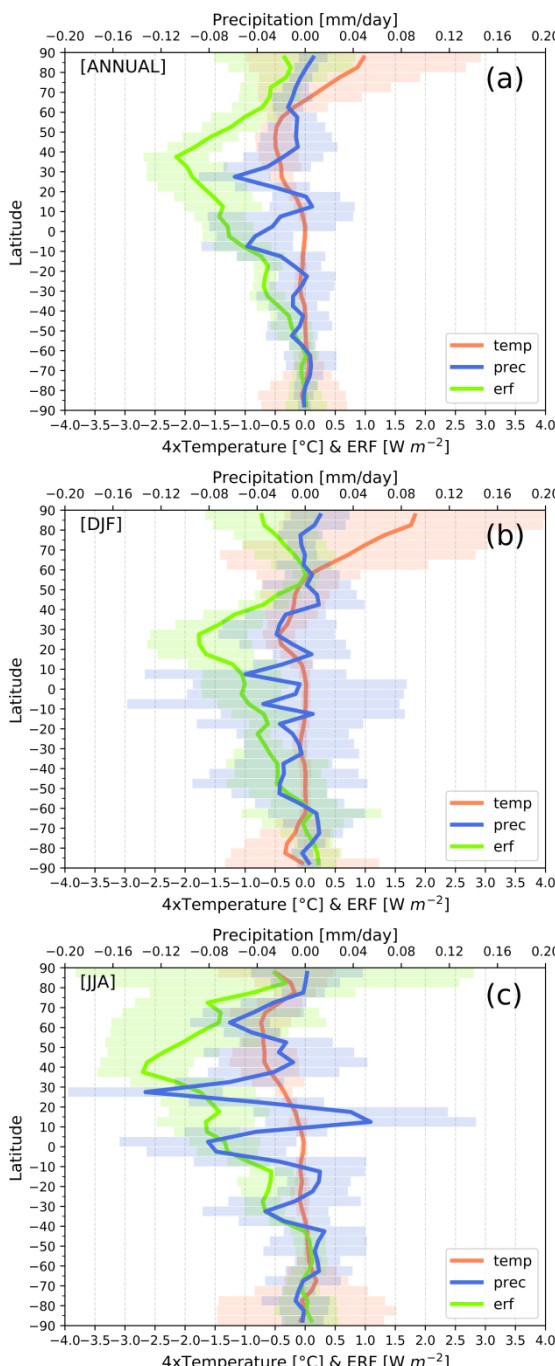

**Figure 4:** Zonal means of the differences between piClim-aer and piClim-control in ERF (W m⁻²) with green line. in near surface temperature (ºC) with pink line and in precipitation (mm/day) with blue line for the ensemble of 10 models on an annual basis (a). for DJF (b) and for JJA (c). The shaded bands show ±1σ of the 10-model ensemble. Mind that the temperature difference has been multiplied by a factor of 4.





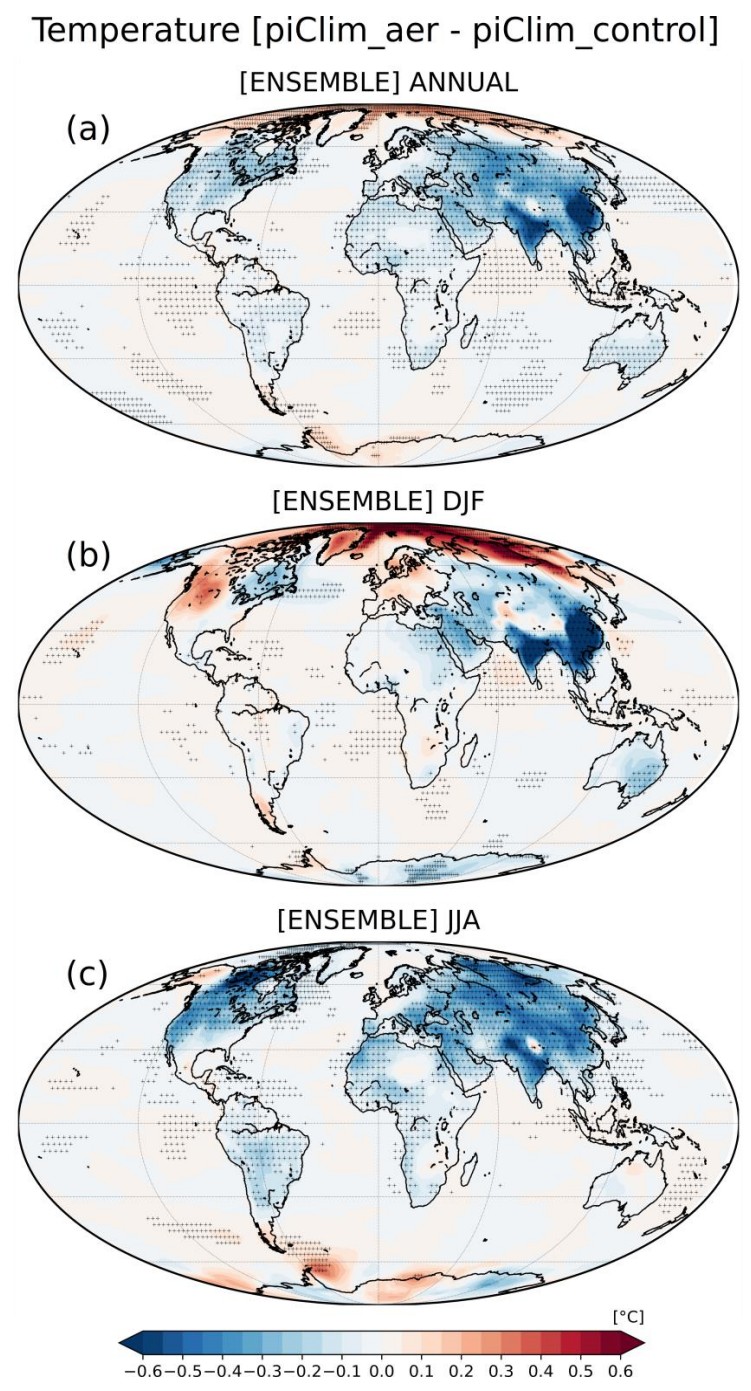

**Figure 5:** Differences between piClim-aer and piClim-control in near surface temperature (°C) for the ensemble of 10 models on an annual basis (a). for DJF (b) and for JJA (c). The dot shading indicates areas in which the differences are statistically significant at the 95% confidence level.





**Figure 6.** Annual differences between piClim-aer and piClim-control in near surface temperature (°C) for each one of the models used for the ensemble. The dot shading indicates areas in which the differences are statistically significant at the 95% confidence level.



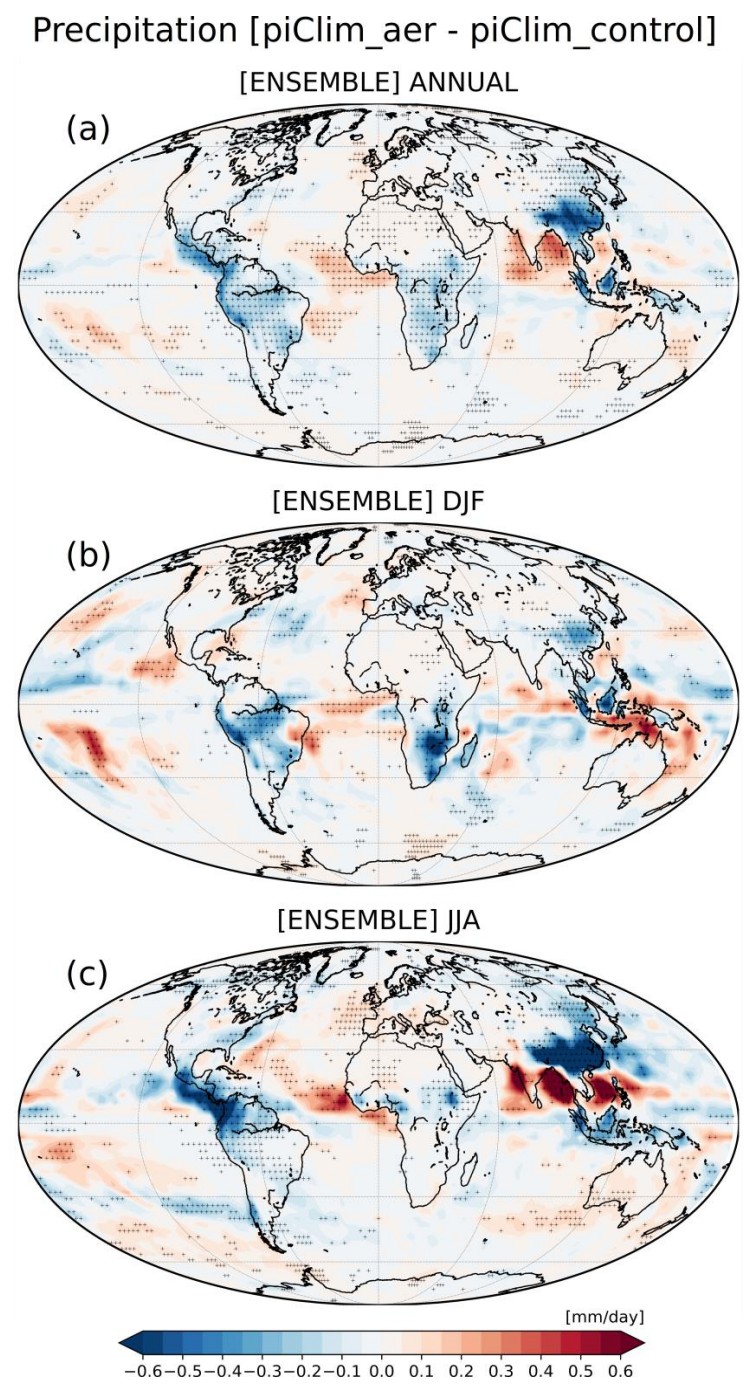

**Figure 7:** Differences between piClim-aer and piClim-control in precipitation (mm/day) for the ensemble of 10 models on an annual basis (a). for DJF (b) and for JJA (c). The dot shading indicates areas in which the differences are statistically significant at the 95% confidence level.







**Figure 8.** Annual differences between piClim-aer and piClim-control in precipitation (mm/day) for each one of the models used for the ensemble. The dot shading indicates areas in which the differences are statistically significant at the 95% confidence level.





## Geopot. Height & Wind [piClim_aer - piClim_control]

**Figure 9:** Differences between piClim-aer and piClim-control in geopotential height (m) and wind vectors at the 850 hPa
pressure level for the ensemble of 10 models on an annual basis (a). for DJF (b) and for JJA (c). The dot shading indicates
areas in which the differences are statistically significant at the 95% confidence level.





## Geopot. Height & Wind ANNUAL [piClim_aer - piClim_control]

**Figure 10.** Annual differences between piClim-aer and piClim-control in geopotential height (gpm) and wind vectors at the 850 hPa pressure level for each one of the models used for the ensemble. The dot shading indicates areas in which the differences are statistically significant at the 95% confidence level. Areas with surface pressure lower than 850 hPa are masked with grey shade.