# Peer review of "Fast responses on pre-industrial climate from present-day aerosols in a CMIP6 multi-model study"

_Atmospheric Chemistry and Physics, 2019_

## Referee Comment (RC1) · Anonymous Referee #1 · 9 Mar 2020

The manuscript by Zanis et al. provides new results from simulations pursued under the AerChemMIP model intercomparison project, which analyse how present-day aerosols may have impacted the global climate system compared to preindustrial times. The focus is on three key variables, i.e. radiative forcing, temperature and precipitation, with some further analysis presented for circulation-related variables. The manuscript is very well written and well within the remit of Atmospheric Chemistry and Physics. The results are not surprising, and mostly confirm the findings of previous studies (with the exception of certain regional details), but they are clearly presented and represent the state-of-the-art of global climate models used in important assessments, therefore the study is a useful addition to the current climate science literature. I do not have any

major objections, and I believe that the paper will be ready for publication following the minor improvements that I describe below.

SPECIFIC COMMENTS:

Page 1, Line 31: I suggest rephrasing to ". . .to shift away from the cooled hemisphere".

Page 2, Lines 30-31: I suggest extending the text in the parenthesis to read like "(affecting climate variables that are mediated by a change in surface temperature and involve the response of the oceans to the forcing)".

Page 3, Line 15: I suggest adding ". . .and variable climate sensitivity per unit aerosol forcing in models" at the end of the sentence.

Page 3, Lines 15-16: The paper by Kasoar et al. (2016) is also a key one when it comes to explaining model diversity in climate responses to aerosols.

Page 3, Lines 17-24: I would say that this paragraph is somewhat out of place here and interrupts the flow. I suggest moving it to a later part of the paper, e.g. at the beginning of the section presenting the ERF results (Sect. 3). In the place of this paragraph in the Introduction, it would be nice to see a small paragraph making it clear what is new in this study. The Introduction jumps a bit too abruptly from a nice summary of aerosol-climate interactions to a brief paragraph of what this paper will present. But a paragraph on e.g. whether some multi-model study like the current one was pursued for CMIP5 or in other single-model studies would be useful. Then followed by a paragraph outlining what the current study adds to what already exists in the literature (i.e. the final paragraph that already exists).

Page 4, Lines 22-23: "from other 3 experiments" -> "from 3 additional experiments".

Page 4, Line 28: I suggest removing "Supporting" as that initially implies to the reader that this refers to the Supplement part of the paper.

Fig. 1: It is never mentioned in Sect. 2 what types of emissions are actually varied in

the sensitivity simulation with present-day aerosol emissions. Does that include only anthropogenic or e.g. also biomass burning emissions. Looking at Fig. 1, it seems that the former is true. But it needs to be clarified.

Page 5, Lines 1-2: Arguably the Middle East has higher emissions than Europe and N. America.

Page 5, Lines 21-23: Yes, that is the most likely (and classic) explanation, but it needs to be supported by a reference or two.

Page 6, Line 6: The numbering/ordering of supplementary figures seems unusual, i.e. Fig. S9 appears in the text after Fig. S2.

Page 6, Lines 6-7: This statement is a bit rushed. The ERF of BC is comparable to (though indeed smaller than) the sulfate forcing locally over the main emission regions (e.g. East and South Asia).

Page 7, Line 1: That's mainly true for DJF, right? If so, please state.

Page 7, Lines 10-11: And what about the other two seasons not shown? Worth mentioning as they may also play a role.

Page 7, Line 9: It is certainly not 'slight' – at its peak it's actually larger than the zonal mean effect in mid-latitudes.

Page 7, Lines 18-20: Is it not relatively easy to look at land snow/ice cover changes in models, or at least at surface albedo changes? This Arctic warming is a quite pronounced feature of this analysis, therefore a more complete explanation would be desirable.

Page 7, Lines 30-31: Suggest rephrasing to "So, the heating due to present-day BC emissions cannot justify this warming in NorESM2".

Page 8, Line 13: Again, why "jump" from Fig. 7 to Fig. 9?

Page 8, Lines 18-19: Please discuss further what the mechanism of this general land drying is thought to be. Is this mostly a thermodynamic effect (due to cooling) or a dynamical effect?

Page 8, Line 23: I do not really see much of an increase anywhere around the tropics in Fig. 4a. Only in JJA.

Page 9, Line 9: I suggest rephrasing to "On an annual basis there is a characteristic dipole pattern of precipitation decreases over East Asia and increases over southern India. . .".

Page 9, Lines 19-20: How do we see a weakening of the monsoon circulation? Please explain a bit more clearly/extensively in the text, as this may not be clear to the reader.

Page 9, Line 32: Please add "fast" between "The" and "response", since Dong et al. (2016) also focused on fast responses.

Page 9, Lines 30-32: Yes, but since the ocean temperatures are kept fixed, the effect of aerosols on the monsoon is only partly realised. Which is fine, given the focus of the paper on fast responses, but it is worth stressing this again here. The studies by Ganguly et al. (2012) and Shawki et al. (2018) provide nice insight into the differing fast and slow effects of aerosols on the South Asian monsoon, as well as the complementary global and regional mechanisms that are at play.

Page 10, Lines 1-2: Does this paper focus on fast or slow responses. Please clarify this (and I recommend that this is done elsewhere in the text too when referencing findings of other papers, given how different fast and slow responses (and mechanisms) can be).

Page 10, Lines 3-4: Most of the papers cited in this sentence cannot be found in the References list of the current manuscript.

Page 10, Line 10: I am not sure I understand: the west African monsoon involves the inflow of moist air from the central Atlantic Ocean into West Africa. What I see in Fig.

9c is more a strengthening than a weakening of the monsoon.

Page 10, Lines 10-12: The study of Hodnebrog et al. (2018) is of relevance when discussing the influence of aerosols on west African rainfall (in that case biomass burning aerosols, but still relevant).

Page 10, Lines 13-15: Yes, but Westervelt et al. (2017) used a coupled ocean-atmosphere model.

Page 11, Lines 25-31: I think the second half of this paragraph needs some tightening/rephrasing.

Page 12, end of Conclusions section: I think here it would be good if the authors could add a little paragraph reminding the reader that all these results were obtained from short-term simulations (and therefore refer to the fast responses), and that the long-term responses will likely be quite different. Also, please mention if a subsequent AerChemMIP study intends to explore aerosol influences on climate on long timescales.

References:

Ganguly, D., Rasch, P. J., Wang, H., & Yoon, J.-H.:. Fast and slow responses of the South Asian monsoon system to anthropogenic aerosols. Geophysical Research Letters, 39, L18804. https://doi.org/10.1029/2012GL053043, 2012.

Hodnebrog, Ø., Myhre, G., Forster, P. M., Sillmann, J., and Sam-set, B. H.: Local biomass burning is a dominant cause of the observed precipitation reduction in southern Africa, Nat. Commun.,7, 11236, https://doi.org/10.1038/ncomms11236, 2016.

Shawki, D., Voulgarakis, A., Chakraborty, A., Kasoar, M., and Srinivasan, J.: The South Asian monsoon response to remote aerosols: Global and regional mechanisms, J. Geophys. Res.,123,11585–11601,https://doi.org/10.1029/2018JD028623, 2018.

[Figure]

2020.

---

## Referee Comment (RC2) · Anonymous Referee #2 · 27 Mar 2020

Review of : Fast responses on pre-industrial climate from present-day aerosols in a CMIP6 multi-modelstudy. By Zanis et al.

This manuscript proposes an analysis of the fast response of climate to anthropogenic aerosol forcing based on CMIP6 models. Although there is no "big surprise" in the results presented and the protocol and analysis is standard ( focusing mostly on seasonal mean response) the main interest of the study is of course to provide an updated view based on state of the art model intercomparison in the context of CMIP6. Some discussions on the response regional patterns are also proposed and interesting. This manuscript is thus relevant to ACP and the AerChemMIP special issue. On the form,

the manuscript and figures are clear and well written. I have nevertheless some questions and comment that could be addressed before publication to ACP.

Section 1.

Fast response vs. slow response discussion. I understand the use of these concepts, especially in view of intercomparing models. Imagine you have to talk to a wider audience interested in the "effective response" of climate to aerosol forcing in a naturally coupled climate system. What can the fast response analysis tell us about that ? I also understand that this concept and related time scale have more meaning on a global scale, but for regional analysis it is not that simple right ? Finally slow response is calculated only via ocean feedback, but there could be also continental reservoir (soil water) with much slower response than atmospheric processes which could induce delayed feedbacks in theory. In some region the oceanic mixed layer could also adjust to radiative perturbation on "intermediate" time scale (between fast and slow response).

Section 2.

Information on the emission sectors should perhaps be a bit discussed. It is not not clear for example if biomass burning emissions are taken or not into account. Are parameterizations of natural emission also enabled ( see my point about dust feedback in the following comments) .

Table 1. There are a couple of models with "no interactive aerosol" , if they also differ from the "prescribed aerosol" category, how can they use the proposed emission scenario. ? Also Is the prescribed climatology consistent with the emission scenario and year used by other models ( which consider year 2014 , i.e. when concentrations had already drastically decreased for some regions compared to the peak of the 70-80's). How are the models dealing with indirect effects (some might have no indirect effects), perhaps that could be a usefull info in the table) ?

Section 3.1.
L20 -24 : I was in fact a bit surprised to see such a positive ERF over the Sahara. I thought that the aerosol mixture (dominated by sulfate as mentioned by the authors , organics, and little bit of BC . . .) would be essentially diffusive enough to stand close to or over the "critical single scattering albedo" as determined by desert albedo : -Is it an effect of only BC (did not seem evident in supplement material) ? -I assume that cloud response contribution to ERF effect might be limited here ( but maybe not for high clouds?). -But also could there be a positive feedback of dust aerosol (more absorbing and usually associated to a positive forcing over desert) contributing to the ERF – in response to dynamical changes ? (assuming of course that ESM account for on-line dust emissions). This "feeling" is reinforced by the result of e.g. CESM2 which clearly show a strong positive ERF signal over well known dust sub regions in Arabia and Thar desert. If such is the case, i.e. if simulated dust burden generally increase , that could be an interesting side conclusion.

An other feature of interest to me was the rather strong negatve ERF change over south-eastern pacific ( along southern America-coast) . How to explain this signal ? Is it a signature of aerosol interaction with low clouds amplifying the aerosol forcing ?

Section 3.2.

The results confirm previous studies. A question perhaps relevant is : has CMIP6 model climate sensitivity also changed with regards to aerosol radiative forcing / emissions compared to CMIP5 models, as is the case for GHG forcing (considering for example possible different cloud responses) ?

Section 3.3

Precipitation response: If we look closely we can notice that over tropical region ( take west Africa fro example) there is a sharp inversion of the signal from land to sea. The precipitation shift due to the large scale differential hemispheric cooling should produce a precip signal more continuous from land to ocean. I think there is in addition also an influence of local surface forcing and response here. Continental surface reacts to

aerosol dimming (less surface flux, enhanced stabilization) whereas only atmospheric absorption is effective over the ocean when SST is kept constant and the precipitation signal becomes positive over the ocean. If you take into account a slight effect of aerosol dimming on SST (e.g. using slab ocean and without considering necessarily a long time scale) you might end up with a negative precipitation signal consistent with land counterpart. This remark perhaps illustrate my earlier concern about the interpretation of "fast response".

There is a robust increase of Indian and SEA monsoon in terms precipitation (in term of wind anomaly it is difficult to see on the figure, but it seems that there is a cyclonic anomaly). However the text is stating a weakening of the monsoon (line 25) linked to southward ITCZ shift. This is a bit confusing. The precip signal obtained is in fact opposite to several studies relating a weakening of Indian monsoon precip due to regional anthropogenic aerosol, in model and observation analyses (as noted by the author a bit later p10 L1-5). Given the importance of this hot spot region, perhaps the author should develop a bit more the analysis of their results here ( an interesting paper could be the one of Bollasina et al., 2014, GRL) ? Could these results be also linked to the forced SST set up as air sea coupling might be particularly important in this region?

Summer precipitation and dynamics over Europe. In term of radiative forcing summer and winter show similar patterns over europe. Intuitively we understand the anticyclonic anomaly generated via regional forcing over Europe for winter, but we would also expect the same for summer ( i.e. regional strengthening of stable conditions by mostly diffusive aerosol). Instead the Icelandic cyclonic anomaly extends over the euro-mediterranean domain associated with increased precipitations. Would that imply that summer aerosol impact over Europe is not driven by regional emissions but rather responds to global scale adjustment to global emissions? Do the authors have some indications that the signal is robust when considering ocean coupling and slow response (from PDRMIP for example) . Looking at model to model variability , it would be good perhaps to have information on effective optical properties ( e.g. total and

absorption AOD, effective singe scattering albedo) for perhaps understanding the sensitivity of response pattern to aerosol parameters.

---

## Author Comment (AC1) · 25 May 2020

We would like to thank Reviewer #1 for the constructive and helpful comments. Reviewer's contribution is recognized in the acknowledgments of the revised manuscript. It follows our response point by point.

1) The Reviewer notes: "Section 1: Fast response vs. slow response discussion. I understand the use of these concepts, especially in view of intercomparing models. Imagine you have to talk to a wider audience interested in the "effective response" of climate to aerosol forcing in a naturally coupled climate system. What can the fast response analysis tell us about that ? I also understand that this concept and related time scale have more meaning on a global scale, but for regional analysis it is not that simple right? Finally slow response is calculated only via ocean feedback, but there could be also continental reservoir (soil water) with much slower response than atmospheric processes which could induce delayed feedbacks in theory. In some region the oceanic mixed layer could also adjust to radiative perturbation on "intermediate" time scale (between fast and slow response)." A nice schematic overview of fast and slow responses concept in precipitation is presented in Fig. SB1 by Myhre et al. (2017) which breaks down the responses for three time scales. An instantaneous forcing due to perturbation of a radiative forcing agent may initially alter precipitation as a result of changes in the atmospheric radiative heating or cooling. The instantaneous change through radiation may further alter the atmospheric temperature, water vapor, and clouds, through rapid adjustments. The instantaneous radiative perturbation and rapid adjustments change precipitation on a fast time scale (from days to a few years) (fast responses). Climate feedback processes through changes in the surface temperature further alter the atmospheric absorption, which occurs on a long time scale (decades) (slow responses). In this way, the climate response to a forcing agent in the fixed SST and sea ice simulations presented in our paper is without any ocean sea ice response to climate change and therefore only weakly coupled to feedback processes, through land surface responses. The role of fast and slow drivers of precipitation changes is species dependent; for BC, fast stabilization effects due to atmospheric absorption can be important even when averaging on long time scales, while for sulfate the slow response dominates in global and zonal means (Samset et al., 2016; Shawki et al., 2018) even though at the regional level the fast response can be also important (Ganguly et al., 2012). Previous studies indicated that the fast precipitation response of BC aerosols dominates over their slow response for global precipitation changes (Andrews et al. 2010; Kvalevåg et al.; Samset et al., 2016; Liu et al., 2018). Ganguly et al. (2012) showed that the precipitation decreases over north-east India and Nepal region are due to the fast response to aerosol forcing based on aerosol emission changes from the preindustrial to present day. The following paragraphs were added

**ACPD**
in the Introduction: " A nice schematic overview of fast and slow responses concept in precipitation is presented in Fig. SB1 by Myhre et al. (2017) which breaks down the responses for three time scales; a) an instantaneous radiative perturbation may initially alter precipitation as a result of changes in the atmospheric radiative heating or cooling; b) the instantaneous change through radiation may further alter the atmospheric temperature, water vapor, and clouds, through rapid adjustments, leading to precipitation change on a time scale from days to a few years (fast responses); c) climate feedback processes through changes in the surface temperature may further alter the atmospheric absorption, which occurs on a long time scale of several decades (slow responses). Under the framework of the Precipitation Driver Response Model Intercomparison Project (PDRMIP), multiple model results indicate that the global fast precipitation response to regional aerosol forcing scales with global atmospheric absorption, and the slow precipitation response scales with global surface temperature response (Myhre et al. 2017; Liu et al., 2018)."

"The role of fast and slow drivers of precipitation changes is species dependent; for BC, fast stabilization effects due to atmospheric absorption can be important even when averaging on long time scales, while for sulfate the slow response dominates in global and zonal means (Samset et al., 2016; Shawki et al., 2018) even though at the regional level the fast response can be also important (Ganguly et al., 2012). Ganguly et al. (2012) showed that the precipitation decreases over north-east India and Nepal region are due to the fast response to aerosol forcing based on aerosol emission changes from the preindustrial era to present day. Previous studies indicated that the fast precipitation response of BC aerosols dominates over their slow response for global precipitation to BC reductions tends to dominate the total response to BC, as also shown in recent PDRMIP results (Samset et al., 2016; Liu et al., 2018)."

2) The Reviewer notes: "Section 2: Information on the emission sectors should perhaps be a bit discussed. It is not clear for example if biomass burning emissions are
taken or not into account. Are parameterizations of natural emission also enabled." The historical CMIP6 input data were used for biomass burning emissions and anthropogenic emissions (van Marle et al., 2017; Hoesly et al., 2018). Natural emissions including dust and sea-salt are calculated interactively by all models following their own parameterizations, except CNRM-CM6-1 which used prescribed fields. The following paragraph was added: " As far as it concerns aerosol-cloud interactions all models include parameterizations for the first and second indirect effects except CNRM-CM6-1, CNRM-ESM2-1, IPSL-CM6A-LR and GISS-E2-1-G that have parameterizations only for the first indirect effect. The historical CMIP6 input data were used for the biomass burning emissions and anthropogenic emissions (van Marle et al., 2017; Hoesly et al., 2018) while natural emissions, including dust and sea-salt, were calculated interactively following their own parameterizations or used prescribed fields based on consistent offline calculations. The model simulations, assigned in Table 1 with "no interactive aerosol", use prescribed aerosol fields which are consistent with the CMIP6 emissions used in the rest of the models. " 3) The Reviewer notes: "Section 2: Table 1. There are a couple of models with "no interactive aerosol", if they also differ from the "prescribed aerosol" category, how can they use the proposed emission scenario." Also Is the prescribed climatology consistent with the emission scenario and year used by other models (which consider year 2014, i.e. when concentrations had already drastically decreased for some regions compared to the peak of the 70-80's). How are the models dealing with indirect effects (some might have no indirect effects), perhaps that could be a usefull info in the table) ?" The models, with no interactive aerosol, use prescribed aerosol fields which are consistent with the CMIP6 emissions used in the rest of the models. In the case of GISS-E2-1-G model, the r1i1p1f1 experiment uses offline aerosol fields that are calculated using exactly the same model version with full chemistry and emissions turned on in the r1i1p3f1 experiment. In the IPSL-CM6A-LR simulations, tropospheric aerosol loads were prescribed as climatological data sets, produced through runs only using the atmospheric (LMDZ), land-surface (ORCHIDEE) and chemistry-aerosols (INCA) components of IPSL-CM using the CMIP6 input data

**ACPD**
(Hoesly et al., 2018; van Marle et al., 2017) for biomass burning emissions and anthropogenic emissions. CNRM-CM6-1 used also prescribed aerosol fields which are consistent with the CMIP6 emissions. We added a paragraph in the revised manuscript as indicated in point 2 (of our response).

4) The Reviewer notes: "Section 2: L20 -24 : I was in fact a bit surprised to see such a positive ERF over the Sahara. I thought that the aerosol mixture (dominated by sulfate as mentioned by the authors, organics, and little bit of BC would be essentially diffusive enough to stand close to or over the "critical single scattering albedo" as determined by desert albedo : -Is it an effect of only BC (did not seem evident in supplement material)? -I assume that cloud response contribution to ERF effect might be limited here (but maybe not for high clouds?). -But also could there be a positive feedback of dust aerosol (more absorbing and usually associated to a positive forcing over desert) contributing to the ERF – in response to dynamical changes? (assuming of course that ESM account for on-line dust emissions). This "feeling" is reinforced by the result of e.g. CESM2 which clearly show a strong positive ERF signal over well known dust sub regions in Arabia and Thar desert. If such is the case, i.e. if simulated dust burden generally increase, that could be an interesting side conclusion." Similarly to our study, positive ERF values over reflective continental surfaces such as desert and polar regions were also detected in previous studies (Shindell et al., 2013; Myhre et al., 2013) and they were attributed to the fact that the very high surface albedo reduces the effect of scattering aerosols, while increasing the effect of absorbing aerosols, leading to a net positive forcing. In the case of CESM2 that shows a strong positive ERF there is an increase of both total AOD and absorbing AOD over the reflective desert regions in piClim-aer - piClim-control which is basically related to both sulfate and BC aerosols. There is however a decrease of total AOD over the sub-Saharan regions and Sahel as well as over the Taklamakan Desert and the desert regions in Australia. Furthermore, we have looked the dust optical depth changes between the two simulations. In part of Sahara desert there is a small increase in dust AOD while over sub-Saharan regions and Sahel there is a small decrease in dust AOD (in agreement with the total AOD

**ACPD**
decrease). In these regions (Sahara and sub-Sahara) it seems that dust changes can contribute to the positive ERF. This is not the case for the Saudi Arabia and Thar deserts where dust AOD seems either to decrease (over Thar) or not changing (over Arabia) which would rather cause a negative (or no change) in ERF at the TOA. Thus the positive ERF values cannot be attributed to dust changes in these regions. Over the Taklamakan Desert and the desert regions in Australia there is a decrease in dust AOD which spatially coincide with the total AOD decreases which could contribute to small negative ERF at the TOA. We added that " Differences in natural aerosols like dust could potentially also contribute to the positive ERF (e.g. in the case of the strong positive ERF in CESM2)."

Figure I: Differences between piClim-aer and piClim-control at annual basis in CESM2 for a) total aerosol optical thickness (550 nm), b) absorbing aerosol optical thickness, c) dust aerosol optical thickness, and d) BC aerosol optical thickness.

5) The Reviewer notes: "Section 2: An other feature of interest to me was the rather strong negatve ERF change over south-eastern pacific ( along southern America-coast) . How to explain this signal ? Is it a signature of aerosol interaction with low clouds amplifying the aerosol forcing ?" This magnitude of the negative ERF signal over south-eastern pacific (along southern America-coast) is not a robust feature among the models although there is a tendency for negative ERF values in most of the models. It is more clearly depicted in CESM2 model and then MRI-ESM2. Aerosol-cloud interactions is a plausible explanation. However a full analysis to decompose the aerosol ERF into components (following Ghan et al. 2013) is beyond the scope of this manuscript but certainly deserves a deeper investigation. It should be noted that for most of the models, all the necessary diagnostics for such a decomposition analysis are not yet available in ESGF. 6) The Reviewer notes: "Section 3.2: The results confirm previous studies. A question perhaps relevant is : has CMIP6 model climate sensitivity also changed with regards to aerosol radiative forcing / emissions compared to CMIP5 models, as is the case for GHG forcing (considering for example possible different cloud
responses) ? The climate sensitivity cannot be fully assessed in the fixed SST runs, but requires fully coupled ocean simulations. Recent work shows that CMIP6 models tend to have a higher range in climate sensitivity than the CMIP5 models (Zelinka et al, 2020). As far as it concerns the comparison of aerosol ERF between CMIP5 and CMIP6 models, we have compared our results (shown in Table 2) with those presented in Allen (2015) and those in Shindell et la. (2013).

We have added the following paragraphs:

"The global annual average of all aerosols ERF (-1±0.24 W m-2) is similar to the multimodel mean ERF value of -0.97±0.43 W m-2 based on 13 CMIP5 models (see Table 1 in Allen, 2015) and the ERF value -1.17±0.29 W m-2 based on 8 ACCMIP models in IPCC AR5 with the patterns being also similar (Shindell et al., 2013)."

"Recent work shows that effective climate sensitivity has increased in CMIP6 models which is primarily due to stronger positive cloud feedbacks from decreasing extratropical low cloud coverage and albedo (Zelinka et al, 2020)."

7) The Reviewer notes: "Section 3.3: Precipitation response: If we look closely we can notice that over tropical region (take west Africa for example) there is a sharp inversion of the signal from land to sea. The precipitation shift due to the large scale differential hemispheric cooling should produce a precip signal more continuous from land to ocean. I think there is in addition also an influence of local surface forcing and response here. Continental surface reacts to aerosol dimming (less surface flux, enhanced stabilization) whereas only atmospheric absorption is effective over the ocean when SST is kept constant and the precipitation signal becomes positive over the ocean. If you take into account a slight effect of aerosol dimming on SST (e.g. using slab ocean and without considering necessarily a long time scale) you might end up with a negative precipitation signal consistent with land counterpart. This remark perhaps illustrate my earlier concern about the interpretation of "fast response". This is an interesting point set by the reviewer which also troubled us for the precipitation increase over the ocean

**ACPD**
close to West Africa. If we look the fields of anomalies in vertical velocities (see Figure II below) we note that the areas of precipitation increase are associated with consistent upward motion anomalies (see the figure below). We also note downward motions over land near the coast. This may imply a local circulation anomaly with upward motion anomaly over the sea, a downward motion anomaly over the land, a low level outward motion anomaly from the land towards the sea and inward motion anomaly from the sea towards the land higher up. If we assume the influence of local surface forcing with more cooling over land than over the ocean (fixed SSTs) (see Figure 6) due to the total aerosol forcing, this may induce a slight local circulation anomaly with more outflow from land towards the ocean consistently with the implied local circulation anomaly. The Sea Level Pressure anomalies (not illustrated in the manuscript) show a small decreasing pressure gradient from land towards the ocean (see Figure III below). Furthermore, the large-scale horizontal wind anomalies at 850 hPa (Figure 9) indicate a westerly anomaly (inward motion from the sea towards the land) which weakens the climatological easterly flow and the outflow of the relatively drier air masses from land towards the ocean.

Figure II: Differences between piClim-aer and piClim-control in vertical velocity (mPa/s) at 850 hPa for the ensemble of 10 models on an annual basis (a). for DJF (b) and for JJA (c). The dot shading indicates areas in which the differences are statistically significant at the 95% confidence level.

Figure III: Differences between piClim-aer and piClim-control in SLP (hPa) for the ensemble of 10 models on an annual basis (a). for DJF (b) and for JJA (c). The dot shading indicates areas in which the differences are statistically significant at the 95% confidence level.

8) The Reviewer notes: "Section 3.3: There is a robust increase of Indian and SEA monsoon in terms precipitation (in term of wind anomaly it is difficult to see on the figure, but it seems that there is a cyclonic anomaly). However the text is stating a weak-ening of the monsoon (line 25) linked to southward ITCZ shift. This is a bit confusing.
The precip signal obtained is in fact opposite to several studies relating a weakening of Indian monsoon precip due to regional anthropogenic aerosol, in model and observation analyses (as noted by the author a bit later p10 L1-5). Given the importance of this hot spot region, perhaps the author should develop a bit more the analysis of their results here (an interesting paper could be the one of Bollasina et al., 2014, GRL) ? Could these results be also linked to the forced SST set up as air sea coupling might be particularly important in this region?

The reviewer is absolutely right. Over East Asia there is an anticyclonic anomaly (Figure 9c) which deteriorates the climatological southerly and southwesterly winds, thus weakening the East Asian monsoon and leading to lower precipitation (Figure 8c). The anticyclonic anomaly indicated by the geopotential height anomaly at 850 hPa over East Asia is also confirmed by a positive sea level pressure anomaly over the region (see Figure III of our repsonse). The importance of this precipitation decrease in East Asia due to the fast response is justified by similar results in the two PDRMIP studies by Samset et al. (2016) and Liu et al. (2018) comparing fast and slow precipitation responses. Both PDRMIP studies show that the fast precipitation response to sulfate aerosols dominates the decrease in south and east Asian precipitation over land while they also reveal an increase over the adjacent oceans. The decrease in land precipitation is consistent with aerosol weakening the land-sea warming contrast leading to anomalous high sea level pressure over land and weakening of the influx of moisture (Monsoon weakening). This is presumably stronger in fixed SST than in ocean coupled simulations because the SSTs cannot also cool in response to the aerosols. With fully coupled runs, both the land and sea cool. And the cooling of the sea mutes some of the impact on the land-sea contrast. So interestingly, the fast response dominates when it comes to the Asian/Indian monsoons. Over India, there is a cyclonic flow anomaly extending from the Arabian Sea towards the Bay of Bengal (Figure 9c) associated with a positive anomaly in precipitation constrained to a latitude lower than 22 deg N (Figure 8c). This cyclonic anomaly reinforces the climatological westerly - southwesterly winds over south India, thus strengthening the Indian monsoon and leading to more
precipitation. However, the cyclonic anomaly weakens the climatological westerly flow at about 22 deg N, thus constraining the positive precipitation anomaly up to this latitude. This is presumably linked with a southward shift of the ITCZ as can be implied by the pattern of positive geopotential height anomaly north of 22 deg N and negative geopotential height anomaly south of 22 deg N (Figure 9c). The circulation changes due to fast responses in Figure 9c shows similarities with the ones presented by Ganguly et al. (2012) (see their Figure 2a) where it is also noted a cyclonic flow anomaly in the Arabian Sea associated with a positive anomaly in precipitation as well as a positive precipitation anomaly over Bay of Bengal. Nevertheless, the JJA fast precipitation response over India is model dependent with a few showing opposite response (Figure S8). We revised our text explaining in more detail our results over this area. Furthermore, we discussed our results in relation to other similar studies pointing also their large sensitivity to the forced SSTs. The following paragraphs were added: "Over East Asia there is an anticyclonic anomaly (Figure 9c) which deteriorates the climatological southerly and southwesterly winds, thus weakening the East Asian monsoon and leading to lower precipitation (Figure 8c). The anticyclonic anomaly indicated by the geopotential height anomaly at 850 hPa over East Asia is also confirmed by a positive sea level pressure anomaly over the region (not shown here). However, the effect of aerosols on the monsoon is only partly realized in these simulations because the ocean temperatures are kept fixed. The importance of this precipitation decrease in East Asia due to the fast response is justified by similar results in two PDRMIP studies by Samset et al. (2016) and Liu et al. (2018) comparing fast and slow precipitation responses. Both PDRMIP studies indicate that the fast precipitation response to sulfate aerosols dominates the decrease in south and east Asian precipitation over land while they also reveal an increase over the adjacent oceans. The decrease in land precipitation is consistent with aerosol weakening the land-sea warming contrast leading to anomalous high sea level pressure over land and weakening of the influx of moisture (Monsoon weakening). This is presumably stronger in fixed SST than in ocean coupled simulations because the SSTs cannot also cool in response to the aerosols

**ACPD**
while the cooling of the sea in the ocean coupled simulations mutes some of the impact on the land-sea contrast. So interestingly, the fast response plays a dominating role for the Asian/Indian monsoons. Over India, there is a cyclonic flow anomaly extending from the Arabian Sea towards the Bay of Bengal (Figure 9c) associated with a positive anomaly in precipitation constrained to a latitude lower than 22 deg N (Figure 8c). This cyclonic anomaly reinforces the climatological westerly - southwesterly winds over south India, thus strengthening the Indian monsoon and leading to more precipitation. However, the cyclonic anomaly weakens the climatological westerly flow at about 22 deg N, thus constraining the positive precipitation anomaly up to this latitude. This is presumably linked with a southward shift of the ITCZ as can be implied by the pattern of positive geopotential height anomaly north of 22 deg N and negative geopotential height anomaly south of 22 deg N (Figure 9c). The circulation changes due to fast responses in Figure 9c shows similarities with the ones presented by Ganguly et al. (2012) (see their Figure 2a) where it is also noted a cyclonic flow anomaly in the Arabian Sea associated with a positive anomaly in precipitation as well as a positive precipitation anomaly over Bay of Bengal. It was shown, however, that the location of the emission region plays an important for shaping the detailed features and magnitude of the response. Decomposition of the total response into fast and slow components indicate that almost all of the precipitation reductions over India (south of 25 oN), Arabian Sea, and Bay of Bengal are a result of the slow response to aerosol forcing, whereas increases in precipitation over the north-western part of the subcontinent as well as decreases over north-east India and Nepal region are due to the fast response to aerosol forcing (Ganguly et al., 2012)."

9) The Reviewer notes: "Section 3.3: Summer precipitation and dynamics over Europe. In term of radiative forcing summer and winter show similar patterns over europe. Intuitively we understand the anticyclonic anomaly generated via regional forcing over Europe for winter, but we would also expect the same for summer (i.e. regional strengthening of stable conditions by mostly diffusive aerosol). Instead the Icelandic cyclonic anomaly extends over the euro-mediterranean domain associated with increased pre-
cipitations. Would that imply that summer aerosol impact over Europe is not driven by regional emissions but rather responds to global scale adjustment to global emissions? Do the authors have some indications that the signal is robust when considering ocean coupling and slow response (from PDRMIP for example). Looking at model to model variability, it would be good perhaps to have information on effective optical properties (e.g. total and absorption AOD, effective singe scattering albedo) for perhaps understanding the sensitivity of response pattern to aerosol parameters." In JJA we have a stronger negative regional ERF over Europe (than in DJF) with the pattern of ERF changes being similar with the pattern of near surface cooling. This points the radiative causes for the near surface cooling. Possibly the level of static stability in JJA is a factor that could play a role for the development of the anticyclonic anomaly. In central and south Europe the lower stability may hinder an anticyclonic anomaly while in north Europe with the higher stability regional forcing promotes the anticyclonic anomaly. However the large scale cyclonic anomaly over the N. Atlantic in JJA that extends towards Europe is presumably linked to global scale circulation adjustment to the global forcing. In DJF we note a negative ERF over Europe which is weaker than in JJA. The negative ERF in DJF does not cause a radiative cooling at near surface. In contrast, we note a slight warming which is dynamically driven from the induced circulation changes, an anticyclonic anomaly over Europe and the cyclonic anomaly over the N. Atlantic. This synoptic pattern anomaly causes warmer air advection and subsidence over Europe. Even though the regional forcing is rather weak in DJF over Europe it could potentially contribute to the development of the anticyclonic anomaly over Europe in DJF. Presumably even in DJF there is contribution from global scale circulation adjustment to the global forcing. The following paragraph was added in the revised manuscript: " In JJA over Europe, the pattern of negative regional ERF anomalies (Figure 3c) is similar to the pattern of near surface temperature anomalies (Figure 6c) pointing to radiative causes for the near surface cooling. At north Europe there is anticyclonic anomaly (Figure 9c) which could be linked to the negative regional radiative forcing and high stability. The large scale cyclonic anomaly over the N. Atlantic in JJA that extends to-

**ACPD**
wards Europe is presumably linked to global scale circulation adjustment to the global scale radiative forcing.

Following the suggestion of the reviewer we added one Figure with information about the ensemble mean difference between piClim-aer and piClim-control simulation for the total AOD and the absorbing AOD. The ensemble mean annual difference between piClim-aer and piClim-control simulations is  $0.027\pm0.012$  for the globe,  $0.046\pm0.020$  for the NH and  $0.011\pm0.003$  for the SH. The following paragraph was also added: "Based on the ensemble of the 10 models on an annual basis, Figure 2 shows, in turn, the differences between piClim-aer and piClim-control for total aerosol optical depth (AOD) and absorbing aerosol optical depth (AAOD) at 550 nm. Their spatial distribution reflects the key emission regions of the anthropogenic scattering and absorbing aerosols. The mean annual difference between piClim-aer and piClim-control simulations for the 10-models ensemble is  $0.027\pm0.012$  for the globe,  $0.046\pm0.020$  for the NH and  $0.011\pm0.003$  for the SH."
Figure 1: Differences between piClim-aer and piClim-control at annual basis in CESM2 for a) total aerosol optical thickness (550 nm), b) absorbing aerosol optical thickness, c) dust aerosol optical thickness, and d) BC aerosol optical thickness.

**Fig. 1.** Figure I:Differences between piClim-aer and piClim-control at annual basis in CESM2 for a) total aerosol optical thickness (550 nm), b) absorbing aerosol optical thickness, c) dust aerosol optical thi
Figure II: Differences between piClim-aer and piClim-control in vertical velocity (mPa/s) at 850 hPa for the ensemble of 10 models on an annual basis (a). for DJF (b) and for JJA (c). The dot shading indicates areas in which the differences are statistically significant at the 95% confidence level. Interactive comment

**Fig. 2.** Figure II:Differences between piClim-aer and piClim-control in vertical velocity (mPa/s) at 850 hPa for the ensemble of 10 models on an annual basis (a). for DJF (b) and for JJA (c).
Figure III: Differences between piClim-aer and piClim-control in SLP (hPa) for the ensemble of 10 models on an annual basis (a). for DJF (b) and for JJA (c). The dot shading indicates areas in which the differences are statistically significant at the 95% confidence level. Interactive comment

Fig. 3. Figure 3:Differences between piClim-aer and piClim-control in SLP (hPa) for the ensemble of 10 models on an annual basis (a). for DJF (b) and for JJA (c).

---

## Author Comment (AC2) · 25 May 2020

We would like to thank Reviewer #2 for the constructive and helpful comments. Reviewer's contribution is recognized in the acknowledgments of the revised manuscript. It follows our response point by point.

1) The Reviewer notes: "Page 1, Line 31: I suggest rephrasing to ": : :to shift away from the cooled hemisphere"." It was revised accordingly as suggested by the reviewer. 2) The Reviewer notes: "Page 2, Lines 30-31: I suggest extending the text in the parenthesis to read like "(affecting climate variables that are mediated by a change in surface temperature and involve the response of the oceans to the forcing)"." It was revised ac-

cordingly as suggested by the reviewer. 3) The Reviewer notes: "Page 3, Line 15: I suggest adding ": : :and variable climate sensitivity per unit aerosol forcing in models" at the end of the sentence." It was revised accordingly as suggested by the reviewer. 4) The Reviewer notes: "Page 3, Lines 15-16: The paper by Kasoar et al. (2016) is also a key one when it comes to explaining model diversity in climate responses to aerosols." The reference of Kasoar et al. (2016) was added in this sentence. 5) The Reviewer notes: " Page 3, Lines 17-24: I would say that this paragraph is somewhat out of place here and interrupts the flow. I suggest moving it to a later part of the paper, e.g. at the beginning of the section presenting the ERF results (Sect. 3). In the place of this paragraph in the Introduction, it would be nice to see a small paragraph making it clear what is new in this study. The Introduction jumps a bit too abruptly from a nice summary of aerosol-climate interactions to a brief paragraph of what this paper will present. But a paragraph on e.g. whether some multi-model study like the current one was pursued for CMIP5 or in other single-model studies would be useful. Then followed by a paragraph outlining what the current study adds to what already exists in the literature (i.e. the final paragraph that already exists)." The paragraph was transferred from Section 1 to Section 2. It was also added a new paragraph as follows: Despite the fact that the slow climate responses of anthropogenic aerosols dominate over the fast responses in zonal means, the fast adjustments are important in regional scale and in global scale for the case BC aerosols as has been noted in several previous single model (e.g. Andrews et al. 2010; Ganguly et al., 2012; Kvalevåg et al. 2013; Li et al., 2018;) and multi-model studies (e.g. Samset et al., 2016; Stjern et al., 2017; Voigt et al., 2017; Liu et al., 2018). 6) The Reviewer notes: "Page 4, Lines 22-23: "from other 3 experiments" -> "from 3 additional experiments"." It was revised accordingly as suggested by the reviewer. 7) The Reviewer notes: "Page 4, Line 28: I suggest removing "Supporting" as that initially implies to the reader that this refers to the Supplement part of the paper." "Supporting" was substituted with "Relevant". 8) The Reviewer notes: "Fig. 1: It is never mentioned in Sect. 2 what types of emissions are actually varied in the sensitivity simulation with present-day aerosol emissions. Does that include only anthropogenic

or e.g. also biomass burning emissions. Looking at Fig. 1, it seems that the former is true. But it needs to be clarified." The sensitivity simulations with present-day (2014) aerosol precursor emissions refer to anthropogenic emissions of SO2, BC and OC. This has been clarified now in the text by adding " ...with emissions for anthropogenic aerosol precursors of SO2, BC and OC set to present-day (2014) levels." 9) The Reviewer notes: "Page 5, Lines 1-2: Arguably the Middle East has higher emissions than Europe and N. America." Middle East was also added in the text among the regions with high SO2 emissions. 10) The Reviewer notes: "Page 5, Lines 21-23: Yes, that is the most likely (and classic) explanation, but it needs to be supported by a reference or two." Two references have been added (Shindell et al., 2013; Myhre et al.,2013). 11) The Reviewer notes: "Page 6, Line 6: The numbering/ordering of supplementary figures seems unusual, i.e. Fig. S9 appears in the text after Fig. S2." Following the reviewer's comment, we re-numbered the supplementary figures according to the order they are discussed in the manuscript. 12) The Reviewer notes: "Page 6, Lines 6-7: This statement is a bit rushed. The ERF of BC is comparable to (though indeed smaller than) the sulfate forcing locally over the main emission regions (e.g. East and South Asia)." In this statement we compare the individual SO2, BC and OC ERF patterns in piClim-SO2, piClim-BC and piClim-OC simulations, respectively, of CNRM-ESM2-1, MRI-ESM2-0 and NorESM2-LM (Figure S3) with the all-aerosol ERF patterns for these models shown in Figure 4. We agree that BC ERF is comparable with SO2 ERF over main emission regions such as East and South Asia as well as in other regions over Africa, Middle East and Indian Ocean. Overall, though, sulfate ERF patterns are similar (not identical) to the all-aerosol ERF patterns indicating the dominating role of sulfates in the all-aerosols ERF. This does not cancel out the role of BC in ERF which for some regions can be even higher than sulfates. We have revised the sentence as follows " ... indicating the overall dominating role of sulfates in the all-aerosols ERF (although there are regions where the role of BC outweighs the role of sulfates in the all-aerosols ERF). 13) The Reviewer notes: "Page 7, Line 1: That's mainly true for DJF, right? If so, please state." This statement refers to Figure 6 for the annual basis, but

it also holds for DJF and JJA. The sentence was revised as follows: However, there are regional differences among the models in the pattern of induced fast temperature responses especially over Europe, North America and Africa on annual basis as well as for DJF and JJA. 14) The Reviewer notes: "Page 7, Lines 10-11: And what about the other two seasons not shown? Worth mentioning as they may also play a role." Of course, the transition seasons SON and MAM may also play a role. However, in order to keep a balance between the discussion and the length of the manuscript we decided to limit the whole discussion throughout the paper in the annual basis and the warm and cold seasons. We hope that this decision is understandable by the reviewer. 15) The Reviewer notes: "Page 7, Line 9: It is certainly not 'slight' – at its peak it's actually larger than the zonal mean effect in mid-latitudes." We agree with the reviewer. The word "slight" was deleted. 16) The Reviewer notes: "Page 7, Lines 18-20: Is it not relatively easy to look at land snow/ice cover changes in models, or at least at surface albedo changes? This Arctic warming is a quite pronounced feature of this analysis, therefore a more complete explanation would be desirable." It should be noted that sea ice does not change in these simulations. So, it is only snow over land or on the sea ice that can lead to a positive snow/ice albedo feedback in these simulations. We have looked the respective changes in snow cover fraction over land between piClim-aer and piClim-control simulations (see Figure IV below). The Figure does not show significant changes over the northern polar latitudes in DJF to justify a positive albedo feedback contribution to the warming signal. This was somehow expected as we have already noted that ERF changes are not consistent with the DJF Arctic Warming and thus Arctic radiation changes does not seem to be a plausible explanation. Furthermore, in these simulations there is no ocean circulation changes which implies as plausible cause for the warming atmospheric circulation changes. This is verified with geopotential height and wind vector changes at 850 hPa. A full quantification of the poleward heat advection is beyond the aims of this study. We have revised the sentence as follows "However, the respective changes in snow cover fraction over land between piClim-aer and piClim-control simulations (not shown) do not support such an albedo feedback.

This is consistent with the fact that ERF changes and thus Arctic radiation changes do not seem to be a plausible explanation for the DJF Arctic Warming. Furthermore, in these simulations there is no ocean circulation changes and it remains as plausible cause for the warming, the atmospheric circulation changes which are verified from the geopotential height and wind vector changes at 850 hPa."

Figure IV: Differences between piClim-aer and piClim-control in snow cover fraction over land for the ensemble of 7 models on an annual basis (a). for DJF (b) and for JJA (c). The dot shading indicates areas in which the differences are statistically significant at the 95% confidence level.

17) The Reviewer notes: "Page 7, Lines 30-31: Suggest rephrasing to "So, the heating due to present-day BC emissions cannot justify this warming in NorESM2"." It was revised accordingly as suggested by the reviewer. 18) The Reviewer notes: "Page 8, Line 13: Again, why "jump" from Fig. 7 to Fig. 9?." Following the reviewer's comment we re-numbered the figures according to the order they are discussed in the manuscript. 19) The Reviewer notes: "Page 8, Lines 18-19: Please discuss further what the mechanism of this general land drying is thought to be. Is this mostly a thermodynamic effect (due to cooling) or a dynamical effect?" In fact, there is not general land drying but rather the reduction of precipitation is seen over parts of continental regions (e.g. East Asia, Central and South Africa, Central and South America). As has been shown in previous studies (Samset et al., 2016; Liu et al., 2018) there is strong correlation between global precipitation "fast" response and atmospheric absorption revealing the thermodynamic influence (due to cooling) on precipitation reduction in the global scale. However, at the regional scales there are dynamical contributions due to circulation changes. Liu et al. (2018) showed that, in sulfate perturbation experiments in ocean coupled simulations (fast+slow responses) or in SST fixed simulations (fast responses), the diabatic radiative term has only a small contribution to the changes in precipitation over almost all regions, whereas regional precipitation is mostly controlled by the atmospheric dynamics (see their Figure S5).This is clear for the case of East

Asia precipitation reduction in monsoon season. The dipole pattern of JJA precipitation responses over East Asia (Figure 8c) is similar to the pattern of fast precipitation responses and of the changes in column-integrated dry static energy flux divergence over this region in perturbation experiments with a five-fold increase in SO4 over Asia (see Figure1 and Figure 7 from Liu et al., 2018). The paragraph was modified as follows: "These fast precipitation responses are characterized generally by a reduction over parts of the continental regions (e.g. East Asia, Central and South Africa, Central and South America) with a global annual change of -0.02±0.01 mm/day. As has been shown in previous studies (Samset et al., 2016; Liu et al., 2018) there is strong correlation between global precipitation "fast" response and atmospheric absorption revealing the thermodynamic influence (due to cooling) on precipitation reduction in the global scale, but regional energy budget analysis clearly indicates the importance of dynamical contributions for heat transport at regional level (Muller and O'Gorman, 2011; Richardson et al., 2016; Liu et al., 2018). Liu et al. (2018) showed that, in sulfate perturbation experiments in ocean coupled simulations (fast+slow responses) or in SST fixed simulations (fast responses), the diabatic radiative term has only a small contribution to the changes in precipitation over almost all regions, whereas regional precipitation is mostly controlled by the atmospheric dynamics either (see their Figure S5)." 20) The Reviewer notes: "Page 8, Line 23: I do not really see much of an increase anywhere around the tropics in Fig. 4a. Only in JJA." Figure 5a shows a decrease of up to -0.05 mm/day peaking at about 7 deg S. In JJA (Figure 5c) this decrease is larger (up to -0.08 mm/day) peaking at around 0 deg. 21) The Reviewer notes: "Page 9, Line 9: I suggest rephrasing to "On an annual basis there is a characteristic dipole pattern of precipitation decreases over East Asia and increases over southern India...".". It was revised accordingly as suggested by the reviewer. 22) The Reviewer notes: "Page 9, Lines 19-20: How do we see a weakening of the monsoon circulation? Please explain a bit more clearly/extensively in the text, as this may not be clear to the reader." Over East Asia there is an anticyclonic anomaly (Figure 9c) which deteriorates the climatological southerly and southwesterly winds, thus weaken-

ing the East Asian monsoon and leading to lower precipitation (Figure 8c). Over India, there is a cyclonic flow anomaly extending from the Arabian Sea towards the Bay of Bengal (Figure 9c) associated with a positive anomaly in precipitation constrained to a latitude lower than 22 deg N (Figure 8c). This cyclonic anomaly reinforces the climatological westerly -southwesterly winds over south India, thus strengthening the Indian monsoon and leading to more precipitation. However, the cyclonic anomaly weakens the climatological westerly flow at about 22 deg N, thus constraining the positive precipitation anomaly up to this latitude. This is presumably linked with a southward shift of the ITCZ as can be implied by the pattern of positive geopotential height anomaly north of 22 deg N and negative geopotential height anomaly south of 22 deg N (Figure 9c). The circulation changes due to fast responses in Ganguly et al. (2012) (see their Figure 2a) shows similarities with our Figure 9c. There is a cyclonic flow anomaly in the Arabian Sea associated with a positive anomaly in precipitation. Also, there is a positive precipitation anomaly over Bay of Bengal in both studies. In our case this precipitation anomaly is more extended (from the Arabian Sea towards Bay of Bengal) because the cyclonic anomaly is also more extended to the east. The following sentence was added: "Over East Asia there is an anticyclonic anomaly (Figure 9c) which deteriorates the climatological southerly and southwesterly winds, thus weakening the East Asian monsoon and leading to lower precipitation (Figure 8c). Over India, there is a cyclonic flow anomaly extending from the Arabian Sea towards the Bay of Bengal (Figure 9c) associated with a positive anomaly in precipitation constrained to a latitude lower than 22 deg N (Figure 8c). This cyclonic anomaly reinforces the climatological westerly - southwesterly winds over south India, thus strengthening the Indian monsoon and leading to more precipitation. However, the cyclonic anomaly weakens the climatological westerly flow at about 22 deg N, thus constraining the positive precipitation anomaly up to this latitude. This is presumably linked with a southward shift of the ITCZ as can be implied by the pattern of positive geopotential height anomaly north of 22 deg N and negative geopotential height anomaly south of 22 deg N (Figure 9c). The circulation changes due to fast responses in Figure 9c shows similarities

with the ones presented by Ganguly et al. (2012) (see their Figure 2a) where it is also noted a cyclonic flow anomaly in the Arabian Sea associated with a positive anomaly in precipitation as well as a positive precipitation anomaly over Bay of Bengal."

23) The Reviewer notes: "Page 9, Line 32: Please add "fast" between "The" and "response", since Dong et al. (2016) also focused on fast responses." It was revised accordingly as suggested by the reviewer. 24) The Reviewer notes: "Page 9, Lines 30-32: Yes, but since the ocean temperatures are kept fixed, the effect of aerosols on the monsoon is only partly realised. Which is fine, given the focus of the paper on fast responses, but it is worth stressing this again here. The studies by Ganguly et al. (2012) and Shawki et al. (2018) provide nice insight into the differing fast and slow effects of aerosols on the South Asian monsoon, as well as the complementary global and regional mechanisms that are at play.." This fact has been pointed in the text as follows: "However, the effect of aerosols on the monsoon is only partly realized because the ocean temperatures are kept fixed. For anthropogenic aerosols, despite the fact that the slow response due to SST change may dominate the total monsoon rainfall and circulation changes over India (Ganguly et al., 2012) and East Asia (Kim et al., 2016), the fast adjustments are important as has been noted in several previous studies. Decomposition of the total response into fast and slow components indicate that almost all of the precipitation reductions over India (south of 25 oN), Arabian Sea, and Bay of Bengal are a result of the slow response to aerosol forcing, whereas increases in precipitation over the north-western part of the subcontinent as well as decreases over north-east India and Nepal region are due to the fast response to aerosol forcing (Ganguly et al., 2012)." It was also added a sentence about the Shawki et al. (2018) results. " Shawki et al. (2018) showed also similar results in the fast precipitation responses, with a precipitation decrease over India and increase over East Asia in JJA (see their Figure S3), due to SO2 reductions (opposite perturbation experiment in relation to our study) in different emission regions. It was shown, however, that the location of the emission region plays an important for shaping the detailed features and magnitude of the response.

25) The Reviewer notes: "Page 10, Lines 1-2: Does this paper focus on fast or slow responses. Please clarify this (and I recommend that this is done elsewhere in the text too when referencing findings of other papers, given how different fast and slow responses (and mechanisms) can be)." In Bartlett et al. (2018) the role of oceanic heat transport is deemed minor because of its slower response time than the time-scale of the analysis (2010-2023). Although the experiments are conducted with a fully coupled model, the analysis time period is relatively short in comparison to the time scale required for large-scale oceanic responses to arise due to the ocean's thermal inertia, especially considering that these are transient experiments with continuously evolving forcings. This implies that, while air-sea interactions are accounted for, atmospheric circulation anomalies induced directly by aerosols will likely play a dominant role in driving the changes discussed thus resembling rather fast responses. This has been clarified in the text. 26) The Reviewer notes: "Page 10, Lines 3-4: Most of the papers cited in this sentence cannot be found in the References list of the current manuscript." The references were added in the reference list. 27) The Reviewer notes: "Page 10, Line 10: I am not sure I understand: the west African monsoon involves the inflow of moist air from the central Atlantic Ocean into West Africa. What I see in Fig. 9c is more a strengthening than a weakening of the monsoon." The slight weakening of the easterlies in JJA takes place over west Sahara. Over the Sahel region there is a weak wind flow (in the ensemble of piclim-control) and indeed as noted the reviewer there is a westerly anomaly mainly over the west coast (Guinea, Sierra Leone, Liberia) which can enhance humidity inflow at this region. However, our comment was mainly referring to the monsoon circulation over the Gulf of Guinea where the flow anomaly is diffluent and this circulation anomaly weakens the monsoon flow. We revised the sentence accordingly as follows: "Specifically, the slight Sahel drying in JJA (Figure 8c) is associated with Sahel cooling (Figure 6c), and in terms of circulation changes with positive GH anomalies and an anticyclonic anomaly presumably weakening the West African monsoon over the Gulf of Guinea (Figure 9c). 28) The Reviewer notes: "Page 10, Lines 10-12: The study of Hodnebrog et al. (2018) is of relevance when discussing the

influence of aerosols on west African rainfall (in that case biomass burning aerosols, but still relevant)." The following sentence was added: Also, local black carbon and organic carbon aerosol emissions from biomass burning activities were suggested to be a main cause of local drying of the atmosphere and the observed decline in southern African dry season precipitation over the last century (Hodnebrog et al., 2016). 29) The Reviewer notes: "Page 10, Lines 13-15: Yes, but Westervelt et al. (2017) used a coupled ocean atmosphere model." This has been specified in the text s follows: In response to U.S. SO2 emission reductions (opposite to the perturbation in our study), in long-term perturbation experiments with three fully coupled chemistry-climate models, a northward shift of the tropical rain belt and the ITCZ was also noted delivering additional wet season rainfall to the Sahel (Westervelt et al., 2017). 30) The Reviewer notes: "Page 11, Lines 25-31: I think the second half of this paragraph needs some tightening/rephrasing." This part of the paragraph was rephrased as follows: NorESM2 is one of the models showing a strong warming in the Arctic in the piClim-aer simulation versus the piClim-control simulation. However, the perturbation experiment piClim-BC with present day BC emissions do not show this warming. Instead, the pattern of Arctic warming seen from the temperature differences between piClim-aer and piClim-control is resembled by the perturbation experiment piClim-SO2 with present day SO2 emissions. 31) The Reviewer notes: "Page 12, end of Conclusions section: I think here it would be good if the authors could add a little paragraph reminding the reader that all these results were obtained from short-term simulations (and therefore refer to the fast responses), and that the long-term responses will likely be quite different. Also, please mention if a subsequent AerChemMIP study intends to explore aerosol influences on climate on long timescales." A sentence was added as follows: Finally, it should be reminded that all the above results are based on 30-year perturbation CMIP6 experiments with fixed SST and sea ice, and hence they refer to fast climate responses through rapid atmospheric adjustments. The slow climate responses in long-term centennial CMIP6 simulations through feedbacks affecting climate variables that are mediated by changes in surface temperature and involve the response of the oceans and

cryosphere to the forcing are in progress within the framework of the IPCC AR6.

[Figure]

[Figure]

**Figure IV: Differences between piClim-aer and piClim-control in snow cover fraction over land for the ensemble of 7 models on an annual basis (a). for DJF (b) and for JJA (c). The dot shading indicates areas in which the differences are statistically significant at the 95% confidence level.**

**Fig. 1.** Figure IV: Differences between piClim-aer and piClim-control in snow cover fraction over land for the ensemble of 7 models on an annual basis (a). for DJF (b) and for JJA (c).